# Proteome profiling of home-sampled dried blood spots reveals proteins of SARS-CoV-2 infections
Claudia Fredolini [1,2], Tea Dodig-Crnković[1,10], Annika Bendes [1,10], Leo Dahl [1,10], Matilda Dale [1,2,10], Vincent Albrecht [1], Cecilia Mattsson[1,2], Cecilia E. Thomas [1], Åsa Torinsson Naluai [3], Magnus Gisslen[4,5,6], Olof Beck[7], Niclas Roxhed [8,9] ✉ & Jochen M. Schwenk [1] ✉

## Abstract

**Background** Self-sampling of dried blood spots (DBS) offers new routes to gather valuable health-related information from the general population. Yet, the utility of using deep proteome profiling from home-sampled DBS to obtain clinically relevant insights about SARS-CoV-2 infections remains largely unexplored.

**Methods** Our study involved 228 individuals from the general Swedish population who used a volumetric DBS sampling device and completed questionnaires at home during spring 2020 and summer 2021. Using multi-analyte COVID-19 serology, we stratified the donors by their response phenotypes, divided them into three study sets, and analyzed 276 proteins by proximity extension assays (PEA). After normalizing the data to account for variances in layman-collected samples, we investigated the association of DBS proteomes with serology and self-reported information.

**Results** Our three studies display highly consistent variance of protein levels and share associations of proteins with sex (e.g., MMP3) and age (e.g., GDF-15). Studying seropositive (IgG$^+$) and seronegative (IgG$^-$) donors from the first pandemic wave reveals a network of proteins reflecting immunity, inflammation, coagulation, and stress response. A comparison of the early-infection phase (IgM$^+$IgG$^-$) with the post-infection phase (IgM$^-$IgG$^+$) indicates several proteins from the respiratory system. In DBS from the later pandemic wave, we find that levels of a virus receptor on B-cells differ between seropositive (IgG$^+$) and seronegative (IgG$^-$) donors.

**Conclusions** Proteome analysis of volumetric self-sampled DBS facilitates precise analysis of clinically relevant proteins, including those secreted into the circulation or found on blood cells, augmenting previous COVID-19 reports with clinical blood collections. Our population surveys support the usefulness of DBS, underscoring the role of timing the sample collection to complement clinical and precision health monitoring initiatives.

## Plain language summary

The COVID-19 pandemic has posed multiple challenges to healthcare systems. A significant gap that remains is a lack of understanding of the impact of SARS-CoV-2 on individuals who did not seek or require hospitalization. To address this, we distribute self-sampling devices to random citizens, aiming to analyze how blood protein levels are affected in people who have had COVID-19 but had no or mild symptoms. Conducting multiple molecular measurements in dried blood, our study confirms clinically known markers and their relationship to infection stages, even if the donors themselves collect the sample. Our work highlights the potential of combining self-sampling with laboratory methods to provide useful information on human health. This convenient patient-centric sampling approach may potentially be useful when studying other diseases.

[1]Department of Protein Science, SciLifeLab, KTH Royal Institute of Technology, 171 65 Solna, Sweden. [2]Affinity Proteomics Unit, SciLifeLab Infrastructure, KTH Royal Institute of Technology, 171 65 Solna, Sweden. [3]Institute of Biomedicine, Sahlgrenska Academy at the University of Gothenburg, 405 30 Gothenburg, Sweden. [4]Department of Infectious Diseases, The Sahlgrenska Academy at University of Gothenburg, 405 30 Gothenburg, Sweden. [5]Sahlgrenska University Hospital, 413 45 Gothenburg, Sweden. [6]Public Health Agency of Sweden, 171 65 Solna, Sweden. [7]Department of Clinical Neuroscience, Karolinska Institutet, 171 77 Stockholm, Sweden. [8]MedTechLabs, BioClinicum, Karolinska University Hospital, 171 64 Solna, Sweden. [9]Department of Micro and Nanosystems, School of Electrical Engineering and Computer Science, KTH Royal Institute of Technology Stockholm, 100 44 Stockholm, Sweden. [10]These authors contributed equally: Tea Dodig-Crnković, Annika Bendes, Leo Dahl, Matilda Dale. ✉e-mail: roxhed@kth.se; jochen.schwenk@scilifelab.se

More than four years since the start of the COVID-19 pandemic, with more than 750 million confirmed cases and nearly seven million deaths, there is still much to learn about a coronavirus infection resulting in a wide range of clinical manifestations. Initially considered a respiratory system disease, the symptoms observed in COVID-19 patients have grown over time, revealing damage to all significant physiological functions, including the cardiovascular, digestive, and nervous systems[1]. A critical factor in COVID-19 pathogenesis is the hyperactivation of the innate immune response with consequent cytokine storm[2-5].

In the first months of the pandemic emergency, the focus was to understand and treat severe conditions, identify effective therapies, and reduce mortality. Indeed, significant progress has been made in understanding the molecular mechanism behind the severe disease in the critically ill and vaccine development; however, little is known about the long-term effects in those with mild or asymptomatic forms of COVID-19. Symptoms such as severe fatigue, memory lapses, and cardiovascular problems have been found in mildly affected patients, especially when symptoms last longer and hinder recovery[6]. It has also been shown that even asymptomatic and mild symptomatic infection may be associated with subclinical lung abnormalities[7,8]. Population-based studies to better understand the heterogeneous phenotypes and genetic and environmental factors associated with disease risk and mortality, long-term effects on an individual's well-being, and to identify therapies that address the molecular diversity are consequently needed[9,10]. Large-scale population studies have been initiated, but an inclusion bias may hamper the generalization of the results. The practical challenge involves including a general population of undiagnosed or non-hospitalized individuals (affected by mild or no symptoms).

One possibility of including a more comprehensive range of phenotypes is using home-sampled dried blood spots (DBS). This strategy can facilitate sample collection across hard-to-reach population groups and reduce the risk of bias in the study design[11]. DBS sample collection has been used since the '60 s, such as for newborn screening, and DBS has been studied with different proteomics techniques[12-15]. Advantages of DBS collection compared to traditional blood sampling include (i) not requiring direct contact and expertise of healthcare personnel, (ii) avoiding traveling and visiting a healthcare center; (iii) representing a convenient format for storage and transportation, and (iv) reduced cost both from a societal and healthcare perspective[16]. The hematocrit bias and imprecision of the collected blood volumes have been issues that hindered even broader use of DBS in medical practice. However, new microfluidic-based DBS devices have recently overcome these two drawbacks to load precise volumes of blood[17-21]. Such volumetric DBS sampling devices can reduce differences in sample quality. This enables profiling and time-resolved monitoring of disease markers collected at an individual's home. Eventually, such tools can allow an accurate and rapid assessment of patients during specific times of the day, at the onset or peak of symptoms, or when seasonal exposures are expected to influence a person's health.

We recently observed an accumulated seroprevalence of 10% in our previous COVID-19 DBS study[22]. Together with available questionnaire data, the study prompted whether it is possible to define additional molecular features of seropositive status in DBS samples. In clinical plasma and serum samples, in-depth proteomic analysis has already delivered valuable insights into the pathology and pathogenesis of COVID-19[23-25]. Our DBS study aimed to demonstrate the utility of self-sampling and identify circulating proteins associated with SARS-CoV-2 infections by considering the serological phenotypes.

With DBS samples collected from random households in the general population in Sweden, we compared seropositive with seronegative subjects (study 1) and donors classified into the early or post-infection phases (study 2) from the first wave of the pandemic. We also studied seropositive and seronegative subjects from the third wave of the pandemic who were not vaccinated at sampling (study 3). For each study, we chose individuals reporting congruent self-reporting symptoms and profiled 276 circulating proteins involved in cardiovascular disease and metabolism using proximity extension assays (PEA). By studying infection-associated profiles in DBS, we confirmed known infection-associated proteins and showed that multiple biological processes are linked with different clinical manifestations of SARS-CoV-2 infections. Analyzing samples collected in a random population will strengthen our understanding of the molecular effects of viral infections and health-related consequences.

## Methods
### Samples and sampling
Per the supplier's instructions, capillary blood samples were obtained by finger-pricking and applying blood droplets onto a quantitative DBS sampling card (qDBS, Capitainer AB, Stockholm, Sweden). The qDBS cards were stored at room temperature until heat treatment before extracting the blood-filled discs.

To compare DBS and EDTA plasma, venous and capillary blood samples were collected from volunteers ($N = 12$) at a healthcare center in the Stockholm region, as previously described[22]. In short, venous blood was collected through venipuncture into EDTA blood collection tubes (K2E K2EDTA Vacuette tube, #454410, Lot# A19104MX, Greiner Bio-One). The tube was centrifuged, and the blood plasma was collected and stored at −20 °C until further use.

In the population studies, capillary blood samples from the general population were obtained by mailing home-sampling kits (MM20-009-01, Capitainer AB, Sweden) to the participants. In April 2020, as previously described[22], kits and a questionnaire were sent to 2000 randomly selected individuals (20-74 years old) in metropolitan Stockholm (Tables 1, 2). In May 2021, kits and a questionnaire were sent to 2000 randomly selected individuals (18–70 years old) in metropolitan Stockholm and Gothenburg

**Table 1 | Demographics of seropositive and seronegative subjects collected in 2020 (study 1)**

| | Seropositive (IgM⁺IgG⁺) | Seronegative (IgM⁻IgG⁻) | *P*-value |
|---|---|---|---|
| Sample numbers | **44** | **37** | |
| **Sex** | | | |
| Female | 25 (56.8%) | 20 (54.1%) | 0.826 |
| Male | 19 (43.2%) | 17 (45.9%) | |
| **Age groups** | | | |
| 20–29 | 14 (31.8%) | 13 (35.1%) | 1 |
| 30–39 | 8 (18.2%) | 7 (18.9%) | |
| 40–49 | 8 (18.2%) | 6 (16.2%) | |
| 50–59 | 5 (11.4%) | 4 (10.8%) | |
| 60–69 | 8 (18.2%) | 6 (16.2%) | |
| 70–74 | 1 (2.3%) | 1 (2.7%) | |
| **Flu-like symptoms** | | | |
| No | 4 (9.1%) | 2 (5.4%) | 0.417 |
| Yes, fever | 19 (43.2%) | 19 (51.4%) | |
| Yes, mild | 12 (27.3%) | 13 (35.1%) | |
| Yes, severe | 3 (6.8%) | 0 (0%) | |
| Missing | 6 (13.6%) | 3 (8.1%) | |
| **Respiratory symptoms** | | | |
| No | 27 (61.4%) | 20 (54.1%) | 0.918 |
| Yes | 4 (9.1%) | 5 (13.5%) | |
| Coughing | 5 (11.4%) | 4 (10.8%) | |
| Difficulty breathing | 5 (11.4%) | 4 (10.8%) | |
| Both | 3 (6.8%) | 4 (10.8%) | |
| Missing | 0 (%) | 0 (%) | |

*As confirmed by Fisher's exact test result.

## Table 2 | Demographics of early-phase and post-phase subjects (study 2)

|  | Early phase, IgM$^+$IgG$^-$ | Late phase, IgM$^-$IgG$^+$ | P-value |
|---|---|---|---|
| Sample numbers | **27** | **43** |  |
| Sex |  |  |  |
| Female | 14 (51.9%) | 28 (65.1%) | 0.782 |
| Male | 8 (29.6%) | 13 (30.2%) |  |
| Missing | 5 (18.5%) | 2 (4.7%) |  |
| Age groups |  |  |  |
| 20–29 | 3 (11.1%) | 6 (14.0%) | 0.432 |
| 30–39 | 6 (22.2%) | 3 (7.0%) |  |
| 40–49 | 5 (18.5%) | 9 (20.9%) |  |
| 50–59 | 3 (11.1%) | 10 (23.3%) |  |
| 60–69 | 4 (14.8%) | 10 (23.3%) |  |
| 70–74 | 1 (3.7%) | 3 (7.0%) |  |
| Missing | 5 (18.5%) | 2 (4.7%) |  |
| Flu-like symptoms |  |  |  |
| No | 9 (33.3%) | 12 (27.9%) | 0.279 |
| Yes, fever | 1 (3.7%) | 6 (14.0%) |  |
| Yes, mild | 6 (22.2%) | 14 (32.6%) |  |
| Yes, severe | 1 (3.7%) | 0 (0%) |  |
| Missing | 10 (37.0%) | 11 (25.6%) |  |
| Respiratory symptoms |  |  |  |
| No | 15 (55.6%) | 28 (65.1%) | 0.842 |
| Yes | 5 (18.5%) | 7 (16.3%) |  |
| Coughing | 1 (3.7%) | 2 (4.7%) |  |
| Difficulty breathing | 0 (0%) | 2 (4.7%) |  |
| Both | 0 (0%) | 2 (4.7%) |  |
| Missing | 6 (22.2%) | 2 (4.7%) |  |

*As confirmed by Fisher's exact test result.

## Table 3 | Demographics of seropositive and seronegative (vaccination-naïve) subjects collected in 2021 (study 3)

|  | Seropositive IgG$^+$ | Seronegative IgG$^-$ | P-value |
|---|---|---|---|
| Samples | **37** | **40** |  |
| Sex |  |  |  |
| Female | 20 (54.1%) | 22 (55.0%) | 1 |
| Male | 17 (45.9%) | 18 (45.0%) |  |
| Age groups |  |  |  |
| 18–29 | 8 (21.6%) | 17 (42.5%) | 0.071 |
| 30–39 | 8 (21.6%) | 12 (30.0%) |  |
| 40–49 | 12 (32.4%) | 4 (10.0%) |  |
| 50–59 | 7 (18.9%) | 5 (12.5%) |  |
| 60–69 | 2 (5.4%) | 2 (5.0%) |  |
| Sensing symptoms |  |  |  |
| None | 13 (35.1%) | 35 (87.5%) | <0.001 |
| Loss of smell | 6 (16.2%) | 0 (0%) |  |
| Loss of taste | 3 (8.1%) | 1 (2.5%) |  |
| Loss of both | 15 (40.5%) | 4 (10.0%) |  |
| Missing | 0 (%) | 0 (%) |  |

*As confirmed by Fisher's exact test result.

### Multi-analyte serology

The serology assays followed a previously described protocol[22] in which proteins were covalently attached to magnetic color-coded beads (MagPlex, Luminex Corp.) using NHS/EDC chemistry. The individual bead populations were then combined into one antigen bead array. Eluates from DBS were diluted by a factor of 2.5 in the assay buffer. The diluted samples (35 µl) were then mixed with 5 µl of the antigen bead array, incubated for 60 min at 23 °C, followed by washing of the beads. The bead-bound human antibodies were then detected using anti-human IgG-R-PE or anti-human IgM-R-PE. The beads were analyzed using a FlexMap instrument (Luminex Corp). The binding data was presented as median fluorescence intensity (MFI) values per antigen and sample, with at least 32 events per bead ID collected for each data point. The MFI data was then processed using a linear model that accounts for unspecific binding.

Serological classification of subjects in studies 1 and 2 was conducted as part of a previous investigation[22]. The subjects for study 1 were identified by multi-analyte classification of anti-S and anti-N titers for IgG and IgM by applying Uniform Manifold Approximation and Projection for Dimension Reduction (UMAP). For study 2, a threshold was set at 6x SD over the peak of the population density to deem a donor seropositive based on anti-S antibody titers for IgG and IgM. For study 3, the 6x SD threshold was used for anti-S and anti-N antibody titers for IgG to classify donors as seropositive or seronegative.

### Affinity proteomics assays

After serology, the residual volumes were used to perform PEAs at the SciLifeLab Affinity Proteomics Unit in Stockholm using Olink panels Cardiovascular III (Product No 95611, Lot No B01116), Metabolism (Product # 95340, Lot # B01109), and Cardiometabolic (Product No 95360, Lot No B02504) according to manufacturer's instructions (Olink Proteomics AB). In brief, EDTA plasma samples were diluted according to the manufacturer's instructions, depending on the panel, 1:100, 1:20, or 1:2025. Based on the assumption that 10 µl of whole blood contains 50–60% fluid, meaning 5–6 µl of plasma, and that we prepared eluates from a starting volume of 100 µl elution buffer, we estimated that our eluates correspond to a plasma sample diluted 1:20. Eluates were diluted at 1:5, 1:1, or 1:101, respectively. For each Olink panel, samples were incubated with 92 pairs of oligonucleotide-labeled antibodies simultaneously. Upon target

(Table 3). Individuals who volunteered to participate in the study were asked to perform self-sampling according to the instructions and return the filled sampling card, questionnaire, and consent form by regular mail. All cards were barcoded and stored at room temperature until use or as stated otherwise. A longitudinal analysis included a volunteer with a PCR-confirmed SARS-CoV-2 infection who collected DBS on five occasions during weeks 2 to 5 after symptom onset in the early recovery phase.

All blood donors gave informed documented consent. The studies were approved by the regional ethical board (EPN Stockholm, Dnr 2015/867-31/1) and the Swedish Ethical Review Authority (EPM, Dnr 2020-01500 and 2021-01106).

### Sample preparation

DBS eluates were prepared to initially determine SARS-CoV-2 serostatus via multiple antigens as previously described[22]. In short, the blood sampling cards were inactivated in an oven (UN55m, Memmert GmbH) at 56 °C for 60 min before ejecting the discs into separate wells of a flat bottom 96-well plate (#734-2327, VWR). The content of the discs was eluted using 100 µl of PBS with 0.05% Tween20 (#97062-332, VWR) and protease inhibitor cocktail (#04693116001, Roche) followed by gentle shaking (170 rpm) for one hour at room temperature. The plates were then centrifuged for 3 min at 3000 rpm (2095 rcf, Allegra X-12R, Beckman Coulter Inc.), and 70 µl supernatant was transferred into a PCR plate (#732-4828, VWR). Sample eluates were stored at −20 °C until analysis.

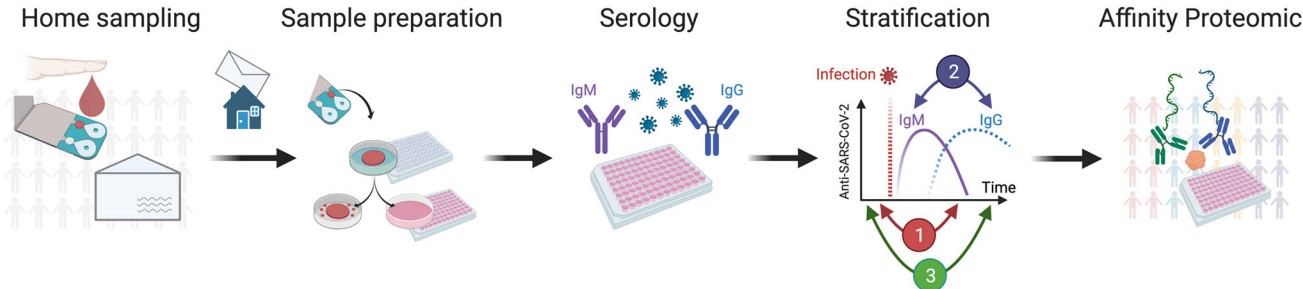

**Fig. 1 | Proteomic profiling of population-based dried blood spot samples.** Home-sampling devices were mailed to random individuals in metropolitan Stockholm and Gothenburg. Dried blood spots (DBS) were collected by finger pricking and mailed back to our laboratory for analysis. We eluted proteins from the DBS discs to first determine antibodies against SARS-CoV-2. Three studies were designed with donors stratified by serostatus and matched on self-reported information: Study 1 from 2020 compared antibody-negative (IgM⁻IgG⁻) with antibody-positive subjects (IgM⁺IgG⁺); Study 2 from 2020 compared IgM-positive (IgM⁺IgG⁻) with IgG-positive donors (IgM⁻IgG⁺); Study 3 from 2021 investigated vaccination-naïve donors who were either antibody-negative (IgG⁻) or antibody-positive (IgG⁺). Proximity extension assays (PEA) were applied to measure the levels of 276 proteins and evaluate their association with the different immune response groups.

recognition, the oligonucleotides in the antibody pairs are brought in proximity allowing for hybridization and DNA polymerization. Reporter sequences were quantified using a microfluidic real-time PCR instrument (Biomark HD, Fluidigm), and data were processed using the software NPX Manager (v.2.1.0.224 and v2.2.0.288, Olink Proteomics AB). The protein levels are reported as semi-quantitative Normalized Protein eXpression (NPX) values. NPX values are calculated from the RT-PCRs Ct values and normalized using the Olink NPX manager software to minimize intra- and inter-assay variation. Reproducibility was judged using samples from two individuals, and technical triplicates of each sample were assayed. These were merged into a single sample per individual by taking the mean across triplicates for each protein measurement. The NPX values are presented as arbitrary units (AU) in the log2 scale.

## Statistics and reproducibility
All data analyses and visualizations were performed using R version 3.6.0[26]. For comparing plasma and DBS, paired two-tailed t-tests were used with the *t.test* function from the *stats* R package (v3.6.0). The *p.adjust* function from the same R package was used to calculate the FDR values. For the global analyses, 264/276 proteins delivered data above the detection limit for >50% of all samples in each study set.

For the population studies, the data were processed as follows: NPX data from two experiments (studies 1 and 2) were bridge-normalized. Bridge normalization was conducted by: (i). Calculate the difference between each protein per paired bridge sample; (ii) For each protein, take the median of these differences across the bridge samples; (iii) Adjust each protein with the median in one of the experimental batches. Each data set was normalized per protein panel using AbsPQN to reduce sample-to-sample variation[27]. Outliers were detected as any sample that in any of the Olink protein panels had a median or IQR ± 3 standard deviations from the mean of each variable. The samples were excluded from the analysis.

Heatmaps were generated using protein-protein correlation values and the *ComplexHeatmap* package (version 2.2.0). The heatmaps represent hierarchical clusters of correlation distances. We used the Gap statistic with 50 bootstrap samples to select the optimal number of protein clusters. Cluster stability was evaluated by calculating the mean Jaccard index (MJI) by bootstrapping proteins of the protein-protein correlation matrices 50 times.

Associations with age and sex were tested using ordinal logistic regression (since ages were coded as age groups in the questionnaire) and logistic regression, respectively, using the *glm* function of the *stats* package (version 3.6.0). To account for protein associations with age and sex, linear regression was used to adjust the data. The residuals of the linear models for each protein were then used to perform the association tests. All *P*-values were FDR adjusted for multiple testing corrections using the Benjamini-Hochberg method and combined using Fisher's method. Tests for

associations between serostatus and self-reported questionnaire variables (sex, age group, and symptoms) used Fisher's exact test (*fisher.test* function of the *stats* package). Unless specified otherwise, the correlation analysis used Spearman's $r_s$ (rho) statistic. Correlations between the determined protein levels and IgG or IgM antibody levels detected against S, RBD, and N antigens were determine in each of the three population sample sets using the *corr.test* function of the *psych* R package (version 1.9.12.31). Two-tailed *P*-values were adjusted for multiple corrections using the Benjamini-Hochberg method. To compare the variance between groups, Levene's test was conducted using the *leveneTest* function of the *car* R package using median-centered data. Comparative analysis with least absolute shrinkage and selection operator (LASSO) regression used the R package *glmnet* (version 4.1) for penalized logistic regression on scaled and centered data to find informative features for serostatus in each data set. The regression was performed 10 times for each data set, and the intersection of the chosen features across the 10 regressions was selected as the informative features. The STRING database version 12 default settings were used to search for interactions between the multiple protein features[28].

To group proteins by temporal trends, SOTA clustering of serology and proteomics data from longitudinal samples was performed using the *sota* function of the *clValid* package (version 0.7). The data were scaled and centered prior to clustering.

## Reporting summary
Further information on research design is available in the Nature Portfolio Reporting Summary linked to this article.

## Results
Using a volumetric microfluidic-based DBS device that collects precisely 10 μl of whole blood, a protocol was tailored to analyze 276 proteins by proximity extension assays (PEA) (Fig. 1). After benchmarking the procedure in a pilot study against paired EDTA plasma samples, DBS collected in Stockholm during the spring of 2020 and in Stockholm and Gothenburg during May of 2021 were analyzed for proteins associated with SARS-CoV-2 seropositivity. The studies revealed proteins relevant to COVID-19 pathogenesis and immune response.

### A comparison of plasma and DBS supports the suitability for deep proteome analysis
To assess the suitability of the DBS preparation for proteomics analyses, protein profiles of 92 circulating proteins related to cardiovascular diseases were investigated (Fig. 2a–c). The levels, correlations, and interquartile range (IQR) between proteins were compared between EDTA plasma collected by venous blood draw and corresponding DBS samples collected at the same visit by finger-pricking from 12 donors (Supplementary Data 1). It was found that 91 out of 92 proteins were detected in > 90% of the sample

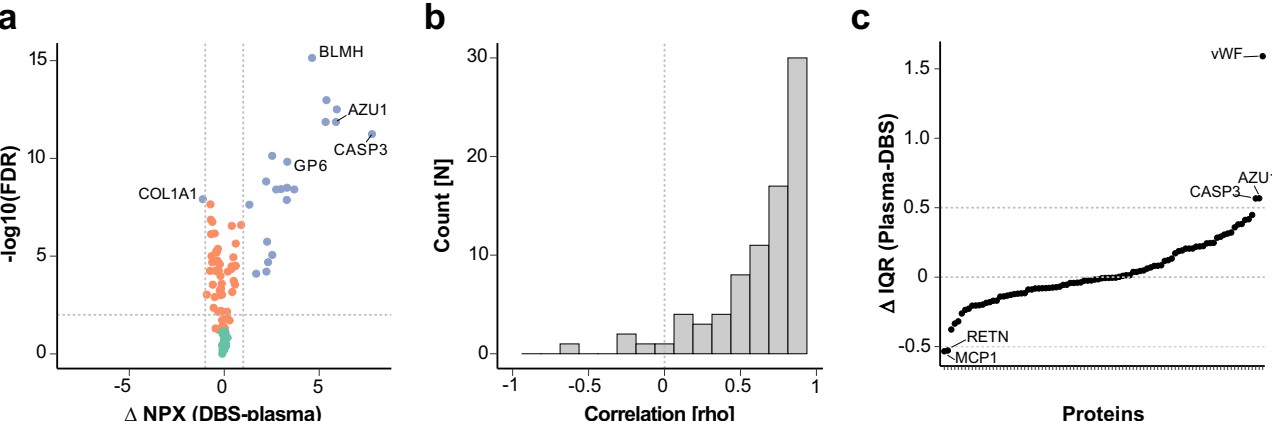

**Fig. 2 | Comparison of dried blood spot and plasma samples. a** The volcano plot displays the difference in relative protein levels between dried blood spot (DBS) and EDTA plasma obtained from 12 donors. The differences in the abundance of 92 proteins, reported as normalized protein expression (NPX), are categorized by FDR < 0.01 (horizontal dotted line) and ΔNPX of ± 1 (vertical dotted lines). Blue dots represent proteins with the most significant differences, orange dots show those with noticeable differences, and green dots represent proteins for which no differences were observed. **b** Frequency of Spearman correlation coefficients for the 92 proteins. The vertical dotted line indicates $r_s = 0$. **c** Differences in protein IQR between DBS and plasma. The vertical dotted lines have been added for orientation at ΔIQR = 0 and of ±0.5.

types, respectively, the investigated proteins could be measured in DBS and paired plasma samples.

As shown in Fig. 2a, paired *t*-tests showed that proteins with elevated NPX abundance levels in DBS (FDR *P* < 0.01) were the platelet glycoprotein VI (GP6, expressed in skin or macrophages), the bleomycin hydrolase (BLMH, expressed in skin keratinocytes), azurocidin 1 (AZU1, expressed in neutrophils), as well as caspase 3 (CASP3, expressed in granulocytes). Likewise, collagen type I alpha 1 chain (COL1A1, expressed in fibroblasts) was more abundant in the plasma samples (Supplementary Data 1). The examples suggest that finger-prick DBS samples can offer improved detectability for skin and blood cell-related proteins for the PEA and possibly other assays.

We also correlated the protein profiles to compare the ranking of the paired samples ($r_s = 0.67$ [−0.61, 0.99]); see Fig. 2b. In general, 62% (57/92) of the protein profiles correlate between plasma and DBS ($r_s > 0.7$). Profiles of cellular proteins such as previously mentioned CASP3, proteinase 3 (PRTN3, expressed in neutrophils), F11 receptor (JAMA, expressed in epithelial cells), and selectin P (SELP, expressed on fibroblasts), were the most discordant ($r_s < 0$). On the other hand, secreted proteins such as NPPB, IGFBP1, CD163, CPB1, and proteins known to leak into blood, such as EPCAM, LDLR, and SELE, were highly concordant ($r_s > 0.95$). Profiles of proteins elevated in plasma agreed with DBS profiles ($r_s = 0.81$ [0.45, 0.99]). The observed discordance between the two specimens was primarily found for proteins with higher levels in DBS samples. In addition, we examined the IQRs of the 92 proteins in the paired DBS and plasma. The IQR of the endothelial coagulation protein VWF was noticeably higher in plasma (alongside AZU1 and CASP3). The proteins MCP1 and RETN, both secreted by hematopoietic blood cells, revealed higher IQRs in DBS (Fig. 2c). Considering all targets, the protein IQRs were not significantly different between DBS and plasma ($P = 0.44$). Testing how the different ranges of detected proteins varied within a given sample type showed that the sample IQRs for DBS were significantly larger than those for plasma IQRs ($P < 1.8 \times 10^{-8}$).

Finally, we investigated the sample types concerning the blood cell expression and protein secretion using differences in NPX (ΔNPX) and correlation ($r_s$) values. With data from the Human Protein Atlas, we annotated the 92 proteins for their RNA expression in tissue[29] and blood cells[30] and the locations of protein secretion[31]; see Supplementary Data 1. We found that 30% of the proteins were not expressed in blood cells. The levels of these proteins were similar between DBS and plasma (ΔNPX = 0.0) and correlated well ($r_s = 0.80$); see Supplementary Table 1. The remaining 70% contained proteins expressed by different blood cell types. The NPX levels of

these proteins were generally higher in DBS than in plasma (ΔNPX = 1.1 [0.4, 2.0]), and the correlation was lower ($r_s = 0.59$ [0.45, 0.72]). As shown in Supplementary Table 2, the proteins secreted primarily into blood were more similar between DBS and plasma (ΔNPX = 0.3; $r_s = 0.71$; N = 20) than proteins secreted to other locations (ΔNPX = 1.3; $r_s = 0.61$; N = 20), or the cellular proteins (ΔNPX = 0.8; $r_s = 0.61$; N = 20). This analysis suggests that protein leakage from blood cells contributed to the differences between the two sample types. Proteins secreted into the circulation by other organs than blood were more similar between the sample types.

The comparative analysis of paired DBS and plasma samples, exemplified here by 92 proteins, revealed differences and commonalities between the sample types. This points to the opportunity to uncover novel associations with DBS and suggests being cautious when aiming to validate findings with the other sample type.

**Population-derived DBS samples for proteome analysis**

In April 2020, we sent 2000 home sampling kits to the Stockholm population to measure antibodies against SARS-CoV-2 in dried blood[22]. The levels of IgM or IgG were determined using multiplexed bead-based assays that included multiple proteins representing the viral antigens. A population-based density cut-off of the antibody levels detected for the coronavirus spike and nucleocapsid proteins was used to classify the serostatus of each sample. Since not all individuals were diagnosed by PCR or experienced symptoms from the infection, we had only self-reported information about a diagnosed infection in one of the studies. For the other, we used only IgM and IgG to group participants into phases post-infection, as suggested by others[32]. In May 2021, a few months after vaccines against COVID-19 became available, we repeated the sample collection by sending a second set of 2000 home-sampling kits to populations in Stockholm and Gothenburg to determine the serostatus during the second year of the pandemic. Using their serostatuses, we selected representative subsets from our collections (N = 228) to perform protein profiling by PEA.

The first study (study 1) from April 2020 was collected during the pandemic's first wave. It consisted of 83 DBS donors, among which 44 participants were selected based on their serological immune response (IgM⁺IgG⁺). These seropositive participants presented the peak of the immune response, which we determined by detecting IgG and IgM against multiple SARS-CoV-2 antigens. The group was matched with 37 seronegative individuals (IgM⁻IgG⁻) based on demographic traits and reported symptoms. There were no significant differences in self-reported symptoms, and only three subjects in the seropositive group reported severe

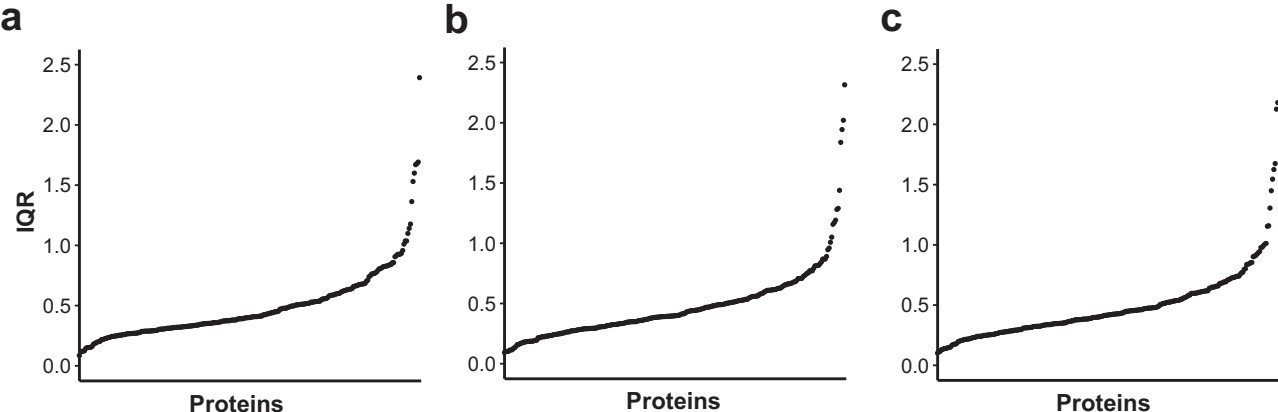

**Fig. 3 | Interquartile ranges of protein levels.** Distribution of protein level variance across dried blood spot (DBS) samples from the population (**a**) study 1 ($N = 81$), (**b**) study 2 ($N = 63$) and (**c**) study 3 ($N = 77$). Each dot represents the interquartile ranges (IQR) of one protein, ranked by the dispersion of normalized protein expression (NPX) values. Proteins with narrow distributions are ranked to the left, and proteins with varying levels are ranked on the right.

symptoms (Table 1). The seropositive subjects of study 1 had only been exposed to the wild-type variant.

The second study (study 2), also collected in April 2020, included 66 participants representing the different phases of the serological immune response against the viral infection. The stratification was based on antibodies detected against the S proteins of SARS-CoV-2. We selected 26 individuals with signs of an acute immune response against the virus by being IgM seropositive only (IgM$^+$IgG$^-$). This group was compared with 40 individuals without detectable IgM levels but being seropositive for IgG (IgM$^-$IgG$^+$). The IgG$^+$ group, annotated as having already passed the acute phase, was slightly older, but otherwise, there were no significant differences between the demographics and the reported symptoms (Table 2). The subjects in study 2 had only been exposed to the wild-type variant.

The third study (study 3) was conducted in late spring 2021 and included 80 unvaccinated participants who donated DBS samples more than a year into the pandemic. We stratified these as seropositive (IgG$^+$) or negative (IgG$^-$) based on antibodies detected against the S and N proteins of SARS-CoV-2. On average, the 37 seropositive individuals reported being infected five months before DBS sampling. Compared to the previous studies conducted during the dominance of the wild-type variant, study 3 represents a set of individuals with much longer possible exposure to different viral strains before the Omicron wave. The groups were matched for sex and age. There was a slight difference in age distribution between the groups, with the seropositive slightly older. The frequency of self-reported symptoms differed, with about a third of asymptomatic seropositive donors (Table 3). The infections of seropositive participants in study 3 could have been caused by different SARS-CoV-2 variants.

**Evaluation of proteomic data in population-derived DBS samples**
In the following, we provide a general overview of the data and then discuss the details and analyses conducted for the three studies. We first evaluated the data globally, searched for possible outliers, and studied the variance of the circulating protein levels in each set to judge the quality and similarity between the data sets. We then determined the common associations of the DBS proteomes with the self-reported traits of age and sex. Lastly, we applied multivariate analysis to identify combinations of proteins to differentiate the serostatus groups, and univariate analysis for associations with symptoms, serostatus, and antibody levels. In each study set, we profiled 276 proteins associated with cardiovascular and metabolic processes such as angiogenesis, blood vessel morphogenesis, inflammation, and cell adhesion.

To begin with, we investigated the general properties of the proteomics data without considering the serostatus categories. Our analysis of the DBS eluates revealed that 260 proteins (94.2%) could be detected in > 90% of the samples from all three study sets. For the downstream analysis, we included 264 proteins (95.6%) above the detection limit for at least 50% of the samples

in all three study sets. Replicated analysis of five unique DBS eluates revealed a high reproducibility of the protein measurements, with > 90% of the proteins reporting a coefficient of variation (CV) < 10% (Supplementary Data 2). Global and unsupervised data analyses were performed to determine the integrity of the data and identify any patterns or biases due to seropositivity. The median NPX and IQR values were used to systematically identify possible outliers by setting the threshold to ± 3 SDs from the mean for each variable. We considered it unlikely that age, sex, symptoms, or serostatus would alter the protein content of samples for the analyzed targets to the degree that identifying a sample as an outlier would have a physiological reason. To account for non-biological differences between DBS samples provided by untrained individuals, we apply the antibody-specific probabilistic quotient normalization (AbsPQN), which we previously developed for affinity proteomic studies of plasma samples[27]. Applying AbsPQN to the three panels used in the three study sets decreased the percent variance explained by the first principal component (PC1) from 40.8% ±15.8% to 15.0% ±1.2%. AbsPQN reduced the differences in the average and distribution of NPX levels. Consequently, AbsPQN-processed data was used to reidentify outliers and for all the downstream analyses. We found eight samples that deviated (Supplementary Fig. 1), thus resulting in their exclusion from the summary tables (Tables 1–3). Out of 236 donors, the proteomics data from 228 samples (97%) qualified for the investigations.

Next, we evaluated the general variation in protein levels to identify stable and highly variable ones. As illustrated in Fig. 3, all data sets presented a similar distribution of IQR values. There was a very good agreement of the IQR values between the three sets ($r_s$ > 0.86, CV = 15%; see Supplementary Data 2). To highlight a few, the most dispersed levels (IQR > 1.5) were found for primarily secreted proteins IGFBP1, MBL2, MEP1B, and SSC4D. Interestingly, MBL2, a protein involved in complement activation, has been previously associated with COVID-19 severity and mortality in intensive care patients[25,33,34]. Among the least variable proteins (IQR < 0.15) were the intracellular proteins CRKL, SOD1, and BLMH, all expressed by various organs. BLMH, a protein highly expressed by the skin tissue[29] and one of the proteins most differentially abundant when comparing DBS with plasma (see above). The observed concordance in IQR values of independent sample sets supported the quality and utility of the data for further detailed analyses of the COVID-19-related phenotypes.

To learn more about the general structure of the data, we conducted unsupervised correlation analyses of 264 protein levels within each of the four serostatus groups. As depicted in the heatmaps presented in the Fig. 4a–c, the overall relationships between the protein correlations differed between the serostatus groups. The distributions of the correlation values centered around zero (Supplementary Fig. 2). A stability analysis of the clusters was performed to prioritize the most stable clusters and choose representative protein correlations across all groups. Cluster #4 of the IgG$^+$

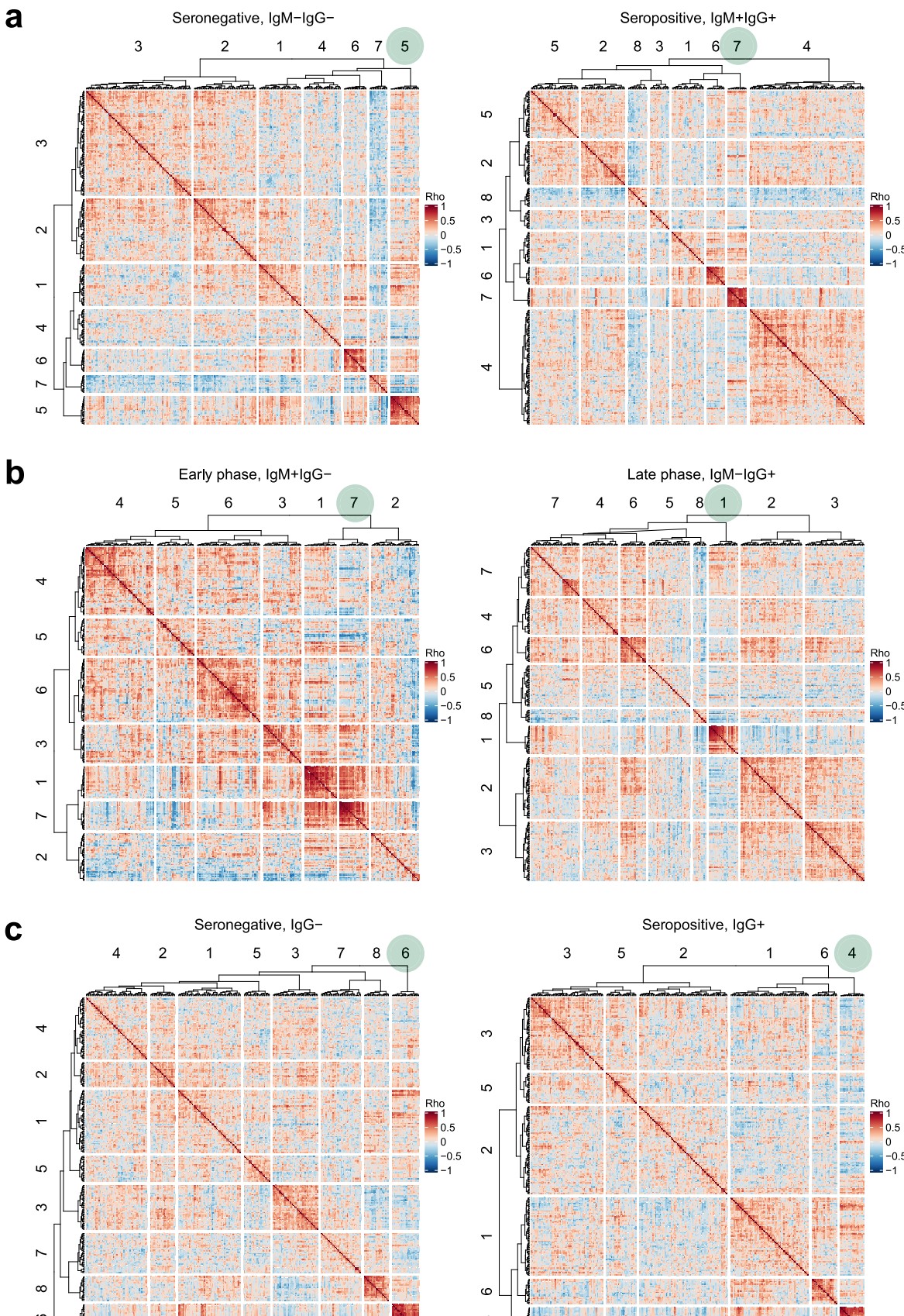

**Fig. 4 | Landscape of protein-protein correlations in phases of the immune response.** The heat maps reveal the inter-protein correlations obtained from hierarchical clustering for the four serostatus groups from (**a**) study 1, (**b**) study 2, and (**c**) study 3. The green circles indicate the clusters containing twelve proteins that grouped together in all sample sets. The number of branches was selected based on Gap statistics.

in study 3 was deemed the most stable cluster with a mean Jaccard index (MJI) = 0.54. The cluster contained 20 proteins originating from different PEA panels. Twelve of the 20 proteins (60%) also clustered in the other five sample sets, and the included ANXA1, PGLYRP1, ITGAM, PLAUR, RETN, TNFRSF10C, NADK, CHI3L1, LCN2, S100P, DEFA1, and PAG1. Interestingly, these twelve proteins originated from the hematopoietic system, including the bone marrow, neutrophils, eosinophils, monocytes, or lymphoid tissue. Despite belonging to different PEA panels, proteins such as LCN2, S100P, PAG1, and PLAUR were shown to correlate highly ($r_s > 0.8$) in all six sample sets. Further details about cluster assignment and protein-protein corrections across study sets can be found in Supplementary Data 3. The cluster analysis suggests that proteome profiling of DBS samples can provide insights into coordinated cellular regulations of the humoral and inflammatory immune response.

## Circulating proteins associated with age and sex

The current knowledge about DBS-derived protein-trait associations is still sparse. Before investigating the relationships between protein levels and SARS-CoV-2 infections, we studied their association with age and sex. The two basic demographic parameters are tested for in nearly all biomedical studies, are known to influence the circulating protein levels in serum or plasma samples, and were collected from all participants in our studies. Consequently, replicating the protein-age and protein-sex associations in three study sets would indicate the data's utility. As we have shown when comparing plasma and DBS, however, the differences between the sample types could influence the outcome of the association comparison. We also note that the age distribution of the three sample sets was slightly different, see Tables 1–3. Using a linear model, we determined the protein-trait associations and performed a meta-analysis to rank the proteins by the combined p-values (Supplementary Data 4). Several proteins were associated with age or sex in all three studies with concordant directions of association (Supplementary Fig. 3). This included the well-known sex-specific protein MMP3 (combined $P = 3.2 \times 10^{-11}$), a protease involved in collagen degeneration. MMP3 has been associated with coronary heart disease and acute respiratory distress syndrome[35] and was studied in COVID-19[25]. In addition (combined $P < 10^{-5}$), sex influences the proteins ALCAM and SSC4D, expressed by the parathyroid glands; CNTN1, found in the brain and sex-specific organs; and IGFBP6, a protein highly expressed in the female sex organs. RNA expression studies support the observed associations with sex[29]. For age, we found strong associations with GDF-15 (combined $P = 1 \times 10^{-17}$), a frequently discussed biomarker for aging[36], across all three studies. In addition (combined $P < 10^{-10}$), meta-analysis identified age-associated proteins in all datasets for the secreted neuronal protein MEPE, the lymphoid protein SELL, the endothelial proteins t-PA or the B-cell receptor CR2. The consistency of age and sex associations across all study sets confirms the data quality and supports its utility in analyzing these in the context of COVID-19.

## Proteins associated with SARS-CoV-2 infections

In the following, we highlight the outcomes of investigating changes in protein levels related to SARS-CoV-2 infections. A LASSO regression analysis was used to identify a combination of proteins that differ between the serostatus groups in each of the three studies. Summary statistics and the group-specific protein values (z-scores) can be found in Supplementary Data 4.

For study 1 (Fig. 5a), 19 proteins were selected, of which 17 (90%) had higher levels in the seropositive group. Ranked by their importance score (Fig. 5b), annexin A11 (ANXA11), found in muscle cells and granulocytes, and the low-affinity immunoglobulin gamma Fc region receptor II-a (FCGR2A), also known as CD32A or FcγRII, were most informative. Both proteins had a reduced abundance in the COVID-19 seropositive group. Interestingly, FCGR2A has been described to trigger a cellular response against pathogens and is involved in phagocytosis, and a recent report suggested that these receptors can mediate the infection of monocytes with the virus[37]. Detecting lower levels of FCGR2A could either indicate an

increased SARS-CoV-2-induced clearing of immune cells or reflect reduced access to the receptor's epitopes while internalizing antibody-bound pathogens. In addition, significant differences were observed for the previously introduced MBL2 and MMP3 and proteins related to different physiological mechanisms. These included proteins secreted by the liver during the stress response and angiogenesis (ANG), a brain- and B-cell-derived neurogenic protein (CHL1), a protease secreted by the pancreas (CPB1), a platelet-derived glycoprotein involved in coagulation (GP1BA) as well as a cytokine receptor related to T-cell immunity (IL2RA). These processes have also been described in studies using venous blood draws[23,38]. SDC4, a cell adhesion protein found in the extracellular matrix of the liver, lung, kidney, and T-cells, has been suggested to act like ACE2 and linked to the cellular uptake of the SARS-CoV-2 virus[39] and revealed anti-inflammatory functions in patients with acute pneumonia[40]. As shown in the network in Fig. 5c, physiological relationships between some of the proteins have been suggested for acute phase processes and innate immunity, platelet activation, coagulation, and cellular adhesion. Correlation analysis of protein and IgG or IgM levels revealed only moderate relationships ($r_s < 0.5$, $P < 0.001$; see Supplementary Fig. 4). We observed the strongest correlation between circulating CHL1 and IgM levels reported for anti-RBD ($r_s = 0.46$; $P = 0.00002$) and anti-S ($r_s = 0.38$; $P = 0.001$). Noteworthy were the negative correlations of FCGR2A with anti-RBD ($r_s = -0.38$; $P = 0.0005$) and S ($r_s = -0.32$; $P = 0.004$). This is supported by studies suggesting that FCGRs mediate the uptake of the antibody-coated virus into monocytes, causing the cells to undergo lytic programmed cell death and reduce levels of circulating FCGR2A[37]. MMP3 and IgG levels correlated with anti-S ($r_s = 0.37$; $P = 0.0006$) and anti-RBD ($r_s = 0.35$; $P = 0.003$) in the opposite direction. Similar trends and relationships were determined for MBL2, VWF, GP1BA, and ANG. Univariate logistic regression for serostatus ranked MBL2, ANG, and FCGR2A on top ($P < 0.01$). Finally, we compared the variances of protein levels between the two groups and found that the distribution of SELL levels ($P = 0.009$) was unequal.

For study 2, LASSO selected five proteins, of which LILBR1 and FAM3C were elevated in the group representing the early phase of the infection (Fig. 5d, e). STRING analysis revealed no known interactions between the proteins; however, syndecan 4 (SDC4) overlapped with the proteins selected in study 1. Elevated levels of SDC4 were found for the later phase group and seropositive in study 1. With LILRB1, an immunoglobulin-like receptor found on monocytes, the metabolism-regulating protein FAM3C, the coagulation factor 11 (F11), and the lung protein cathepsin H (CTSH), a variety of biological processes were represented. Interestingly, SDC4, LILRB1, and CTSH share expression in lung tissues. Correlation analysis revealed negative coefficients between IgM levels detected for the S antigen and the levels of the proteins CTSH and SDC4 ($r_s > 0.37$; $P < 0.003$). Using univariate logistic regression, the five proteins were weakly significantly associated with serostatus ($P < 0.03$). When comparing the variance of protein levels in each group, the levels of CCL5 were most unequally distributed ($P = 0.001$).

For study 3 (Fig. 5f), only one protein was selected by LASSO: the complement C3d receptor 2 (CR2), also known as CD21. Found primarily in the lymphatic system and on B-cells, elevated levels of CR2 were associated with prior infection with SARS-CoV-2. Interestingly, CR2 has been described as a human receptor for the Epstein-Barr virus (EBV), representing an additional element of innate immunity and host-virus interactions[41]. There was a positive correlation between levels of CR2 and anti-S antibodies ($r_s = 0.38$; $P = 0.0006$), and when using univariate logistic regression, a more significant association with serostatus than for the markers shortlisted above ($P = 0.0004$). It is worth noting that, compared to study 1, infections of the seropositive participants in study 3 were not limited to the few months at the start of the pandemic. When comparing the protein level variances, the macrophage protein of CCL24 was most unequally distributed ($P = 0.009$).

Finally, we used common health-related information to perform a meta-analysis of the self-reported symptoms. As shown in Tables 1–3, we

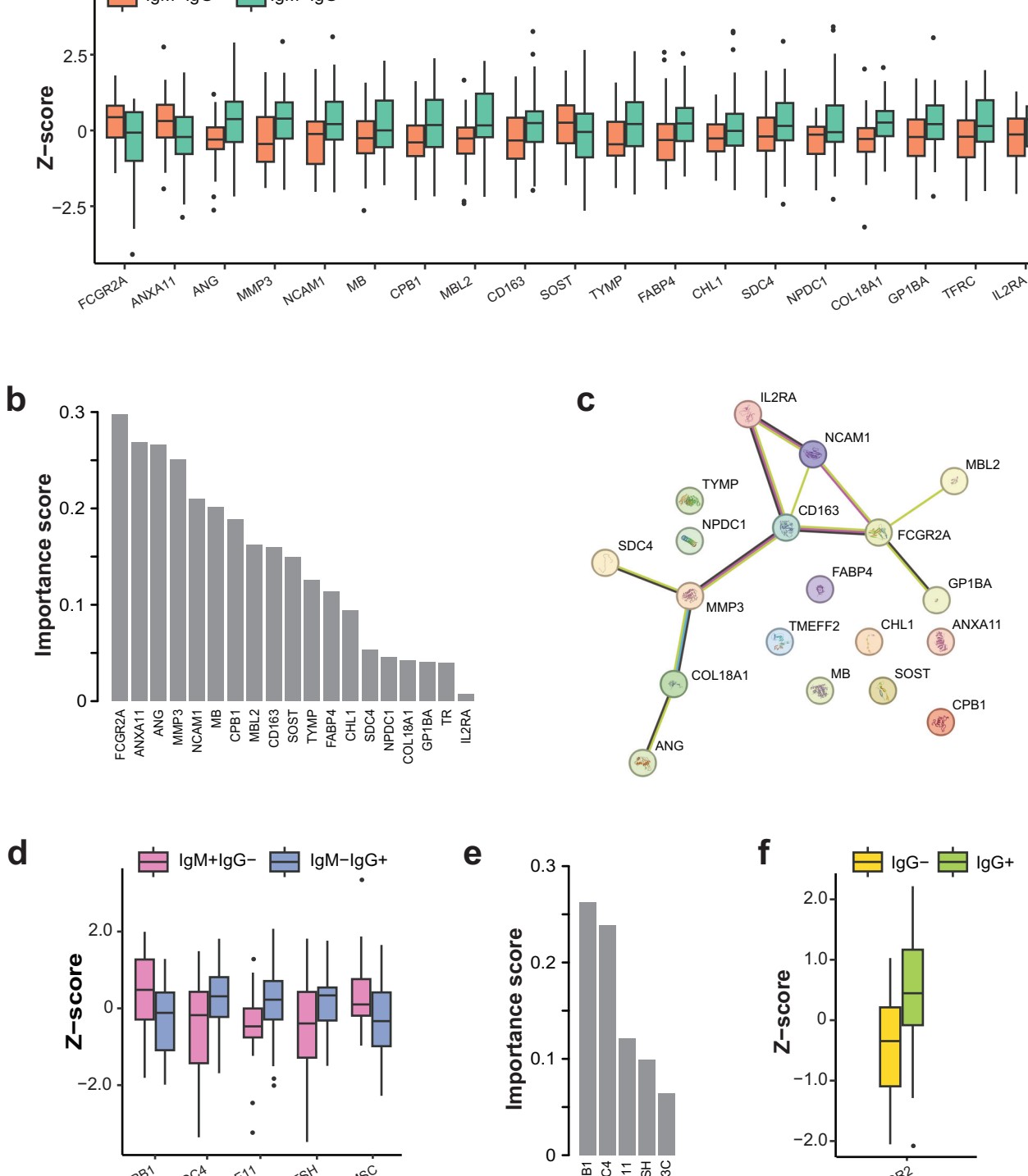

**Fig. 5 | Proteins related to SARS-CoV-2 infection. a** Least absolute shrinkage and selection operator (LASSO) analysis shortlisted proteins differentiating seropositive ($N = 37$) and seronegative ($N = 44$) subjects in study 1. The *y*-axis represents the centered and scaled data provided as normalized protein expression (NPX) values. **b** Ranked importance score of selected proteins. **c** Using the STRING database, we identified interactions between the selected features and obtained a network centered around acute phase processes and innate immunity, host-virus interactions, coagulation, and cellular adhesion. **d** LASSO selected proteins from study 2 comparing donors representing the early ($N = 22$) and late infection phases ($N = 41$), and (**e**) corresponding importance scores. **f** LASSO selected proteins from study 3, comparing seropositive ($N = 40$) and seronegative donors ($N = 37$). The boxplots show the 25% and 75% quantiles (lower and upper hinges) with the median in the center, the whiskers extending to hinges ±1.5× interquartile ranges (IQR), and visible data points outside these ranges.

asked the participants in all three studies for COVID-related symptoms such as fever, breathing difficulties, or loss of taste and smell. Top-ranked (combined $P < 0.01$) were two previously described age-associated proteins (GDF15, SELL) and COL18A1, an extracellular adhesion collagen expressed primarily by the liver and involved in endothelial cell migration, as well as C1QTNF1, a secreted multifunctional adipokine found in smooth muscles and adipose tissue. These associations were less significant and overlapped with those observed for age or sex. No interactions have yet been reported in STRING to suggest a direct connection between their physiological function.

In subjects representing two waves of the pandemic and pre- and post-infection phases, profiling proteins related to cardiometabolic processes in DBS samples revealed insights observed in studies performed in serum or plasma collected in the clinic. Our investigation confirmed coordinated co-regulations of protein levels in immune response, cell adhesion, and cellular virus entry processes.

## Discussion

Circulating proteins are essential sentinels for specific pathway activation and organ status. They allow monitoring disease progression and response to therapy in severe COVID-19 hospitalized cases and investigating the causes behind long-term symptoms experienced by mild COVID-19 patients[4]. Despite most individuals with COVID-19 experiencing only mild to moderate symptoms, others may develop pneumonia, acute respiratory distress syndrome, and, in general, a multi-pathologic and complex clinical picture. The most well-known pathologic features include cytokine release syndrome or "cytokine storm", lung, cardiovascular, and kidney dysfunction, increased thrombotic risk, and down-regulation of adaptive cellular immunity[2-5]. Furthermore, high levels of cardiac biomarkers such as Troponin, BNP, and MBL2 were identified as strong predictors of mortality[25,35].

The COVID-19 pandemic highlighted the need for fast and effective strategies for population health surveillance, monitoring infectious diseases spreading, and investigating the short- and long-term consequences on a heterogeneous population[9]. Thus, new and precise analytical capabilities linked to patient-centric sampling at home, such as dried blood spots (DBS), will become an important contribution[42]. We applied volumetric DBS sampling in the general population in two metropolitan areas in Sweden, and based on serological analysis, we selected 228 individuals representing different serostatuses. We then chose to investigate >250 circulating proteins to capture the effects of a SARS-CoV-2 infection on the proteome level.

First, we benchmarked the approach by comparing finger-prick DBS with plasma from venous blood draws. As predicted, proteins with a higher abundance in DBS samples originated from the skin, intracellular, and blood cells (e.g., GP6, AZU1, CASP3), suggesting cell lysis as the main reason. Variability analysis revealed that cellular proteins such as BLMH and SOD1 are among the least variable proteins (IQR < 0.15). This indicates comparable counts of blood cells trapped in the discs across donors. We consider the presence of blood cells in the DBS sample an inherent advantage for expanded whole-blood profiling. Examples of these observations are the group of twelve proteins that cluster together across all study sets and our ability to detect disease-relevant proteins expressed by circulating monocytes, such as FCGR2A or LILRB1[30]. As reported recently, monocytes can be infected by SARS-CoV-2, leading to cell death[37]. Consequently, proteins expressed by monocytes will be cleared from the circulation and would be detected at lower levels.

Our investigation also describes a streamlined procedure to assess the data quality, to account for variance in sample collection by a layman donor, and to identify anomalies in the data. We introduce antibody-specific probabilistic quotient normalization (AbsPQN), a tool previously used to successfully counterbalance technical factors in other affinity proteomics measurements[27]. AbsPQN reduced the variance of the first principal component (Supplementary Fig. 1) and centered the correlation between (random) pairs of proteins obtained from different PEA panels near zero (Supplementary Fig. 2). Additionally, the variance of protein levels across different data sets was highly concordant (Fig. 3). When testing the

associations of proteins with sex and age, two variables common for all three studies, we observed a clear concordance in the direction of the trends (Supplementary Fig. 3). These findings suggest that AbsPQN is an impactful data processing tool, especially useful for home sampling studies where dried blood is collected remotely and shipped to a central lab.

The studies 1 and 2 presented here focused on understanding the circulating proteins of non-hospitalized individuals during the early phase of the first wave when the wild-type variant was dominant, and neither vaccines nor extensive testing capabilities were available. No participant reported being diagnosed through PCR tests or being vaccinated. In study 3, conducted one year later and between the second and third wave, the Alpha variant dominated, and the Delta variant was on the rise[43]. Compared to studies from the first wave, infections could have occurred during a much longer period of time. According to reported answers, participants got infected, on average, approximately five months before sampling (ranging from 0–12 months). The variants and distribution of infection time points could have contributed to greater differences in the seropositive group's short- or long-term disease activity, thus resulting in fewer significant hits. According to the questionnaire data (Tables 1–3), most participants were asymptomatic or only reported mild symptoms. For such random population studies, the time elapsed between getting infected and donating a blood sample might differ between participants. In studies 1 and 2, the time between infection and sampling was shorter than for study 3. Besides being influenced by how long the self-reported symptoms lasted, the use of medication to counteract the symptoms might have further contributed to the heterogeneity of the donors combined into a respective group. Because older citizens received the vaccines first, the selected vaccine-free subjects for study 3 were generally younger than those for study 1 and 2. Compared to clinical studies that, ideally, collect samples around a narrow peak of the infection, our study investigated random samples with a less-coordinated sample timing; hence, less-distinct significant differences between the assigned groups were to be expected. As further indicated by testing the heterogeneity of variance of protein levels between the groups, we found some discordant distributions for targets expressed by the lymphoid tissues. This could point at differences in current or previous disease activity levels.

One attractive advantage of collecting DBS through micro-sampling is obtaining blood repetitively. To test this proof-of-concept, we obtained samples from a single donor, who collected DBS during the phase recovery (Supplementary Fig. 5). Measured in duplicate, the data hints at longitudinal fluctuations of protein levels during a time when COVID-induced IgM levels decreased, but IgG levels remained stable. This small pilot demonstrated that protein levels change between the time of infection and sampling, as shown for the proteins mentioned in our DBS-based population surveys. Such longitudinal changes can add to the group differences. However, to our current knowledge, this is still the first proteomics survey conducted in a general population that uses self-sampling of dried blood spots. The integrity of the sample type and robustness of the developed workflow support the use of home sampling to generate high-quality molecular data from large cohorts.

The comparison in study 1 highlighted some physiological aspects of COVID-19 pathology. In addition to the proteins highlighted above, MBL2, a protein associated with ICU mortality for COVID-19 patients[25], was found at high levels in individuals with a seropositive group compared to healthy seronegative individuals. The same trend was observed for IL2RA, a protein involved in leukocyte activation and previously associated with prolonging illness in severe COVID-19 patients[44]. Supported by the STRING analysis, the identified blood protein signatures represented cell-mediated immune response and tissue damage, mechanisms lying behind severe COVID-19[23,24]. We also found SDC4 to be a protein associated with the later phase of the infection. Using single-cell data, expression of SDC4 is noticeably elevated in the alveolar and basal respiratory cells, which could reflect the effects of the SARS-CoV-2 infection on the lung. Studying subjects from less-acute phases did not confirm SDC4 but revealed the B-cell receptor CR2. Also known to bind EBV, CR2 was more abundant in seropositive

donors collected a year into the pandemic and could reflect prolonged exposure to the virus.

We successfully used PEA to quantify proteins collected in filter paper-based DBS samples obtained in clinical or professional settings. The technology showed convincing capabilities to detect proteins decades after sampling[15], with a good correlation between DBS and samples collected by venous blood draw[45]. Our application confirms these observations and expands the utility of PEAs as a suitable method for detecting large numbers of proteins in DBS. However, it is well known that intra- and inter-individual variability in hematocrit levels or other blood-cell counts may affect the dispersion of whole blood on paper-based DBS collection matrices. Using a volumetric sampling device guarantees higher precision in the downstream analysis. The device implemented here allows the automation of the extraction procedure, avoids the manual selection of punching areas, and increases the throughput of the analysis. Our data confirmed the robustness and reproducibility of protein quantification (CV < 10%) and a convincing concordance between protein profiles in EDTA plasma collected by venous blood draw and DBS collected by finger pricking ($r_s > 0.7$). This also confirmed that the selected dilutions of the DBS samples provided approximately the same number of proteins detected above the LOD as in a cell-free plasma sample.

An inherent limitation of studying DBS samples is the need for sensitive methods for protein quantification[11]. This study focused on stable proteins occurring at medium to high abundance levels in the blood circulation, which can also be detected after diluting eluates obtained from DBS samples (see Material and Methods). Our subsequent efforts should improve the elution procedures of DBS samples for PEA-based assays to quantify proteins of lower abundance, such as inflammatory cytokines. Others have recently used DBS and other affinity-based methods to detect the well-known IL6 or TNF alpha proteins[46], two cytokines we did not target. Using PEA is an inherent challenge as the reported values are not quantitative concentrations; antibody binding may be influenced by off-target binding, post-translational modifications, protein interactions, and the overall composition of the sample. Efforts comparing PEA or other affinity-based assays with mass spectrometry are emerging; however, a sensitivity gap still reduces the possibility of corroborating each method on all targets[47]. An additional challenge arises from the differences when preparing dried blood samples for proteomics analysis: Affinity-based methods prefer detergent-based extraction buffers compared to the detergent-free samples analyzed by mass spectrometry[13]. Moreover, since the chosen method was built on pre-selected panels of proteins, we could have missed some relevant metabolic and inflammatory markers described in the current COVID-19 literature[48–50]. DBS sampling has been tested with other affinity proteomics methods and targeted MS[51]. The benefit of PEA includes the high-multiplex capacity and the excellent sensitivity (low pg/mL); nevertheless, when quantitative monitoring of biomarkers is needed, other methods such as targeted MS, ELISA, or other quantitative multiplex assay platforms, such as Luminex, Quanterix or MesoScale, would be preferable. According to a comparative analysis[52], intra and inter-assay precision of ELISA (CV 2–12%) or targeted MS (CV 5–11%) was comparable with the values we and others have reported for PEA in DBS samples (CV < 10%; Supplementary Data 2).

Our population samples were collected anonymously from random households; thus, no follow-up of the participating donors was possible. Even though we observed associations between serostatus and proteins, there could be unknown factors, such as BMI, genetics, medication, travel, socio-economic, and lifestyle, contributing to the difference in this analysis. These factors will be necessary to investigate in future studies that expand on the presented results. Looking forward-, other recent studies have shown that the use of longitudinal DBS re-sampling enables the monitoring of short-term changes even in multi-omics analysis[53].

## Data availability
The proteomics and serology data supporting the findings in this study are available at the SciLifeLab Data Repository (https://scilifelab.figshare.com) under the doi identifiers 10.17044/scilifelab.25050422[54] and 10.17044/scilifelab.14555520[55]. The datasets are under restricted access because these represent individual-level human data. As described in the repository, access to the data can be granted for non-commercial validation purposes and upon reasonable request to the corresponding authors. Source data are available as Supplementary Data 5.

## Code availability
Analysis codes used in the study are available at https://github.com/Schwenk-Lab/Olink-DBS[56].

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

## Acknowledgements

First, we thank all anonymous volunteers who contributed blood samples to our study. We thank Mun-Gwan Hong and Benjamin Murrell for the statistical support, and the members of the Nilsson Lab for fruitful discussions. We thank the KTH node of Protein Production Sweden (PPS), a national research infrastructure funded by the Swedish Research Council, the SciLifeLab infrastructure unit for Autoimmunity and Serology Profiling, and the team at Olink Proteomics AB for their support. Figure 1 was created using BioRender.com. The study was supported by grants from the SciLifeLab National COVID-19 Research Program, financed by the Knut and Alice Wallenberg Foundation (2020.0182, 2020.0241); Sweden's innovation agency Vinnova (2020-04451); and SciLifeLab's Pandemic Laboratory Preparedness program (VC-2022-0028). We acknowledge support from the Erling Persson Family Foundation for the project "Future-proofing against COVID-19", and the Knut and Alice Wallenberg Foundation for funding the Human Protein Atlas. This work was partially supported by the Wallenberg AI, Autonomous Systems and Software Program (WASP) funded by the Knut and Alice Wallenberg Foundation.

## Author contributions

A.B., M.D. and C.M. performed experiments. T.D.C., L.D., A.B., V.A. and C.E.T. curated data and performed data analysis. O.B., Å.T.N., M.G., N.R. and J.M.S. conceptualized the study and collected clinical specimens. J.M.S., N.R. and C.F. designed the experimental analyses and supervised the work. C.F., T.D.C. and J.M.S wrote the original draft of the manuscript, and all authors reviewed and edited the manuscript.

## Funding

## Competing interests

O.B. and N.R. are co-founders of Capitainer AB, a company that commercialized the blood collection device for microsampling. J.M.S. has, unrelated to the presented work, received speaker fees from Roche Diagnostics, travel support from Luminex Corp. and Olink AB, and conducted contract research for Luminex Corp. and Capitainer AB. All other authors declare no competing interests.

## Additional information

**Supplementary information** The online version contains Supplementary Material available at https://doi.org/10.1038/s43856-024-00480-4.

