## [Peer Review File · Communications Medicine]

Referee expertise:

Referee #1: Proteomics and COVID

Referee #2: Proteomics and DBS and COVID

Reviewers' comments:

Reviewer #1 (Remarks to the Author):

The authors analysed DBS-derived plasma proteins from patients at different stages of SARS-CoV-2 infection by PEA. Several protein differences were found between the seropositive and seronegative donors. Additional comparison between IgM+ vs IgG+ patients revealed 2 altered inter-protein correlations with complement C2. The authors suggested the DBS can be used for precise proteome analysis with clinical relevance.

The application of DBS in clinical diagnosis is not new. This manuscript used DBS for COVID-19 by combining the volumetric home-sampled DBS with proximity extension assays (PEA). This application is novel, but my main criticisms are as follows:

- 1) The study cohorts are rather small.
- 2) The findings do not reveal much new information compared to previous papers.
- 3) What do you envision as the main application of this DBS-derived profiling?

Other major concerns:

1. How to define IgM+/-, IgG+/-? Please provide more detailed method for the serology.
2. In the pilot study analysing DBS against paired EDTA plasma samples, most increased proteins are from blood cells. Comparing DBS (containing cellular proteins from blood cells) vs EDTA plasma is not a valid comparison. One can only focus on the common plasma proteins detected in both type of samples.
3. There is a need to explain why COL1A1 is reduced in DBS.
4. Please include a supplemental table for all the measured values of the 276 proteins in the 4 groups (if not for each individual samples) and which cluster each protein is assigned to (as shown in Figure 4).
5. Why do you choose Spearman correlation? Even with the Spearman correlation, most of the association with $Rho = \pm 0.3 - \pm 0.4$ are considered low/moderate correlation.
6. The authors claim the DBS proteome profiling reveal valuable molecular insights into protein changes associated with seropositivity for coronavirus. Most of the comparisons, however, do not pass the filter $FDR P < 0.05$.
7. Did you correlate the proteins changes over a time course - progression of COVID-19 infection and recovery?
8. PEA is limited to a few hundred target proteins and biased by the panel selection.
9. I could not find the data for the longitudinal analysis (5 separate occasions during weeks 2 to 5 after symptom onset), which is mentioned in the method.

Minor:

1. Fig.2. $FDR < 0.01$ (*horizontal* dotted line)
2. Dataset S2, column J is not necessary. Column for "Interplate – Median CV %" does not exist.
3. Method section, convert 3000rpm to rcf.

Reviewer #2 (Remarks to the Author):

This is an interesting study, performed in several parts. First, the authors performed a comparison of Olink's Proximity Extension Assay (PEA -- multiplexed ELISA with PCR readout) for 92 proteins in Dried Blood spots (DBS) with blood samples from the same individuals, using DBS sampling kits designed by the authors (a collection device using a fixed volume of blood). As in an earlier study (ref 15), good correlation between the two datasets were found, so the authors continued with large-scale study of DBS on blood samples collected in 2020 from possible COVID-affected patients. These samples were then analyzed by Olink who performed PEA using a panel of 276 proteins (Olink's cardiovascular III and cardiometabolomic panels).

I was not familiar with this PEA technique, but Olink claims that their 2-antibody per protein, with DNA tags to enable PCR amplification of the signal, eliminates the cross-reactivity observed in standard ELISA. The use of panels targeting proteins in specific pathways would make this technique a targeted "semi-quantitative technique" (as stated on page 20 of the manuscript). Presumably, these panels of proteins were selected because the pathways were already known to be associated with COVID. Interestingly, different panels of proteins could distinguish between 4 classes according to different immune responses (IgM-IgG-, IgM+IgG+, IgM+IgG- and IgM-IgG+) (page 9). I am more familiar with mass-spectrometry-based biomarker discovery -- and the rather low percentage of potential biomarkers that actually successfully pass the validation step. Therefore, I am curious as to whether the authors intend to validate any of the potential biomarker proteins that they uncovered, and if so, how would this validation be done?

I was also a bit confused by the discussion of protein "pairs", and how these pairs were selected from all of the up or down-regulated proteins (page 12). Perhaps a bit more explanation would be helpful.

Although the paper in general is well written, it could benefit from a bit of language-polishing by a native speaker. (Some examples are "It is worth noticing" -> It is worth noting (line 282) and "Hard breath" -> Difficulty breathing, on page 28). To assist the authors, I have attached an edited word file.

Coming from a mass-spectrometry background, and knowing the cost of a standard ELISA assay, I am curious about the cost of this Olink PEA analysis for 276 targeted proteins. I am also curious about the cost of Olink versus label-free LC/MS/MS for the same number of proteins detected. Or possibly a comparison of a limited number of Olink targets versus multiple-reaction monitoring MS for biomarker validation.

The amplification capability of PEA would certainly be an advantage (PEA literature claims fg/mL sensitivity), but most of the proteins described in this paper are of medium to high abundance (page 18, lines 436-437). As the authors note on page 18, there is a study that combines PEA with mass spectrometry (ref 43), but according to this reference only 35 proteins were compared between PEA, data-dependent and data-independent LC/MS/MS methods. As the authors note on page 18, this is an important area for future research.

In summary, this is an interesting paper, but PEA is only semi-quantitative, and validation of the results presented here has not been addressed. I also have the feeling that PEA is probably very expensive (2 antibodies per protein, with DNA tags and PCR readout) -- and it seems to be only offered by Olink as a service. This would seem to limit its applicability.

Reviewer #1 (Remarks to the Author):

The authors analysed DBS-derived plasma proteins from patients at different stages of SARS-CoV-2 infection by PEA. Several protein differences were found between the seropositive and seronegative donors. Additional comparison between IgM+ vs IgG+ patients revealed 2 altered inter-protein correlations with complement C2. The authors suggested the DBS can be used for precise proteome analysis with clinical relevance.

The application of DBS in clinical diagnosis is not new. This manuscript used DBS for COVID-19 by combining the volumetric home-sampled DBS with proximity extension assays (PEA). This application is novel, but my main criticisms are as follows:

1) The study cohorts are rather small.

>> We understand the reviewer's concern, but it remains challenging for us to respond appropriately to a relative statement ("rather small"). We agree that many larger studies ($N > 1000$) have been conducted with regular blood samples. However, to our knowledge, DBS is still most often limited to smaller-sized studies ($N < 200$) and does not cover deep proteomic analyses.

Nonetheless, we regard our cohort size as sufficient to demonstrate our approach's utility. Our literature survey did not flag our study as small but medium-sized. We acknowledge the limitation of using fewer samples regarding statistical power and possibilities to conduct training and test analyses.

As requested by the reviewer and editor, we have added a new study set to the manuscript, increasing the total number of samples to ~230.

2) The findings do not reveal much new information compared to previous papers.

>> We understand the reviewer's concern. Still, it remains challenging to respond appropriately when we do not know which publications are being referred to that would contain the information we replicated in our study. Nonetheless, in the revised version we further clarified that the aim was not to postulate new biomarkers. Instead, demonstrated that analysis of DBS samples can provide insights into the disease phenotypes that match the current knowledge about SARS-CoV-2 infections, such as identifying associations of FCGR2A or MBL2.

From our own literature survey, we found one study by Suhre et al. (Ref #48) that summarized the findings from other efforts using serum or plasma samples. Unfortunately, many studies focused on inflammatory proteins, which we have not included in our pilot studies due to their low abundance. Besides lifting some inflammatory proteins, the authors' investigation showed a quite common discordance in findings between available studies. One core differentiator is the study design and inclusion criteria. It is worth noting that most studies today have focused on the severely ill or those who seek medical care. Our studies are population surveys of random subjects representing a wider range of phenotypes.

In essence, our intention was not to postulate new biomarkers but to demonstrate the utility of DBS in revealing clinically discovered biomarkers in population-based sample collections. We have revised the section on page 4 to clarify the aims:

On page 4, we revised the last section of the introduction:

"...Our DBS study aimed to demonstrate the utility of self-sampling and identify circulating proteins associated with SARS-CoV-2 infections by considering the serological phenotypes.

With DBS samples collected from random households in the general population in Sweden, we compared seropositive with seronegative subjects (study 1) and donors classified into the early or post-infection phases (study 2) from the first wave of the pandemic. We also studied seropositive and seronegative subjects from the third wave of the pandemic who were not vaccinated at sampling

(study 3). For each study, we chose individuals reporting congruent self-reporting symptoms and profiled 276 circulating proteins involved in cardiovascular disease and metabolism using proximity extension assays (PEA). By studying infection-associated profiles in DBS, we confirmed known infection-associated proteins and showed that multiple biological processes are linked with different clinical manifestations of SARS-CoV-2 infections. Analyzing samples collected in a random population will strengthen our understanding of the molecular effects of viral infections and health-related consequences..”

3) What do you envision as the main application of this DBS-derived profiling?

>> We thank the reviewer for this relevant question. We have added more on this in the introduction and discussion.

As of today, the main application of our approach will be to simplify the sampling procedures used for molecular analysis. This will enable more frequent and time-resolved health monitoring studies of treatment response or disease progression in high-risk subjects through in-depth and high multiplexed protein profiling. Demonstrating the utility of our approach in random participants who were not trained to provide samples at a given time point supports the potential to link clinical assessments with home sampling schemes.

We are working on several studies that expand on the approach presented here. One example of how the concept of our work can impact future studies has recently been discussed in GenomeWeb (see the last sections): <https://www.genomeweb.com/proteomics-protein-research/human-protein-atlas-using-olink-tech-move-plasma-proteomic-profiling>

Other major concerns:

1. How to define IgM+/-, IgG+/-? Please provide more detailed method for the serology.

>> We thank the reviewer for pointing this out. In the methods sections, we have now added further details about the serology analysis and approach to determine the serostatus. Further details can be found in Ref # 22.

In addition, we state on page 7: *“The levels of IgM or IgG were determined using multiplexed bead-based assays that included multiple proteins representing the viral antigens. A population-based density cut-off of the antibody levels detected for the coronavirus spike and nucleocapsid proteins was used to classify the serostatus of each sample.”*

2. In the pilot study analysing DBS against paired EDTA plasma samples, most increased proteins are from blood cells. Comparing DBS (containing cellular proteins from blood cells) vs EDTA plasma is not a valid comparison. One can only focus on the common plasma proteins detected in both type of samples.

>> We thank the reviewer for the valuable comment. The reason for conducting this analysis was to investigate which proteins would differ in abundance and profiles. For the following, we used information from the Human Protein Atlas to investigate this matter further.

Importantly, we did not select proteins due to prior knowledge about their presence or absence in blood cells but chose to observe post hoc if differences could be explained by location. In fact, about 1/3 of the proteins are not expressed by blood cells; see tables below and in the supplementary (Tab S1-2), and 1/3 are not known to be secreted. Thirdly, plasma proteins can originate from cellular leakage due to cellular turnover, apoptosis, or damage during blood collection and storage.

Stimulated by the reviewer's comment, we conducted further analyses to control our investigation for secretion and blood cell expression. This was added to the revised manuscript version on pages 6-7. In summary, we observed that proteins secreted into blood presented were more similar when judged by lower differences in NPX levels (Δ NPX) and higher correlation than those proteins that were not secreted or primarily secreted into other locations.

Location of secretion	N	Average Δ NPX	StdDev Δ NPX	Average rho	StdDev rho
Not secreted (leakage)	27	0.83	2.14	0.61	0.41
Other main location + Blood	24	1.30	2.08	0.61	0.28
Blood	20	0.26	0.96	0.71	0.26
Intracellular/Membrane	8	0.93	1.49	0.65	0.32
Matrix	6	-0.39	0.40	0.82	0.09
Other locally	4	-0.44	0.20	0.80	0.11
Gastric	2	-0.16	0.11	0.95	0.03
Secreted - no data	1	0.38	-	0.90	-

Proteins not expressed in blood cells had the lowest differences in NPX levels between DBS and plasma (Δ NPX = 0). The profiles of these proteins also had a higher average correlation ($r = 0.8$) than groups with proteins expressed by blood cells ($r \leq 0.7$). Proteins expressed by all blood cell types had the highest level difference (Δ NPX = 2) and lowest agreement between DBS and plasma ($r = 0.5$).

Blood Cell RNA	N	Average Δ NPX	StdDev Δ NPX	Average rho	StdDev rho
Not detected	28	-0.02	0.87	0.80	0.20
Detected in many	23	0.37	1.23	0.72	0.26
Detected in some	17	0.72	1.51	0.63	0.30
Detected in all	14	2.01	2.61	0.45	0.42
Detected in single	10	1.35	2.48	0.55	0.37

Investigating the relationship between secretion and blood cell expression showed that a major fraction of proteins primarily secreted into blood were not found in blood cells. Proteins linked to blood cells primarily were secreted to other locations or not secreted. These observations allowed us to reason that the source for differences between DBS and plasma is, as expected, due to protein leakage from blood cells residing in the DBS sample. However, we also found that proteins secreted from other organs were measured to a high degree of concordance between the sample types.

3. There is a need to explain why COL1A1 is reduced in DBS.

>> We thank the reviewer for this observation. The protein COL1A1 is highly expressed in fibroblasts, smooth muscle, and skin (See RNAseq data below). We expected that COL1A1 should be more abundant in plasma than DBS because of the blood collection process. The needle penetrates several tissue layers, including smooth muscles, to reach a major blood vessel. For blood collection on DBS, however, a lancet does not penetrate deeply into the tissue and should only access capillary blood vessels.

Source: <https://www.proteinatlas.org/ENSG00000108821-COL1A1>

4. Please include a supplemental table for all the measured values of the 276 proteins in the 4 groups (if not for each individual samples) and which cluster each protein is assigned to (as shown in Figure 4).

>> We have followed the reviewer’s suggestion and provided group-specific values for each protein in the supplementary in Data S3. Individual-level data (z-scores) will be deposited on the SciLifeLab’s Data Repository (<https://scilifelab.figshare.com>). As we will describe in the repository, access to the data can be granted in accordance with local guidelines and for validation purposes.

As we mentioned in the introduction to the rebuttal, we used a more streamlined approach to process the data more uniformly. This included applying AbsPQN + outlier detection before data analysis. Consequently, the outcome of the clustering analysis has changed, and the results were updated accordingly.

Compared to the initial submission, only one cluster determined from the correlation analysis of the IgG+ group in study 3 achieved an MJL > 0.6. Following the cluster assignment of these 20 proteins across other study sets, it was revealed that 12 proteins continued to cluster together. Interestingly, these 12 proteins originated from all three PEA panels; hence, independent measurements of the same samples generated their data. Despite being on separate panels, proteins such as LCN2, S100P, PAG1, and PLAUR were shown to correlate highly ($\rho > 0.8$) in all six sample sets. As shown in the STRING network, interactions between these proteins have been suggested.

Target	Panel	UniProt	Study 1, IgM-IgG-	Study 1, IgM+IgG+	Study 2, IgM+IgG-	Study 2, IgM-IgG+	Study 3, IgG-	Study 3, IgG+
ITGAM	CAM	P11215	5	7	7	1	6	4
DEFA1	CAM	P59665	5	7	7	1	6	4
LCN2	CAM	P80188	5	7	7	1	6	4
TNFRSF10C	CVD3	O14798	5	7	7	1	6	4
RETN	CVD3	Q9HD89	5	7	7	1	6	4
PGLYRP1	CVD3	O75594	5	7	7	1	6	4
CHI3L1	CVD3	P36222	5	7	7	1	6	4
PLAUR	CVD3	Q03405	5	7	7	1	6	4
PAG1	MET	Q9NWX8	5	7	7	1	6	4
S100P	MET	P25815	5	7	7	1	6	4
NADK	MET	O95544	5	7	7	1	6	4
ANXA11	MET	P50995	5	7	7	1	6	4

5. Why do you choose Spearman correlation? Even with the Spearman correlation, most of the association with $Rho = \pm 0.3$ - ± 0.4 are considered low/moderate correlation.

>> We thank the reviewer for the comments. We used the ranked-based Spearman method due to its lower sensitivity to outliers than the Pearson correlation.

We agree that the correlations were moderate due to measuring random samples from the general population. Our study groups were only selected due to their antibody levels and to match a corresponding group. We do not have a clinical measure of disease severity, but we can assume that each group is more heterogeneous than those of clinical cohorts. See also our response about p-values below.

6. The authors claim the DBS proteome profiling reveal valuable molecular insights into protein changes associated with seropositivity for coronavirus. Most of the comparisons, however, do not pass the filter $FDR P < 0.05$.

>> The reviewer raises an important point. Highly significant p-values are certainly a very appealing indication of the possible impact of a finding. However, p-value levels alone are less meaningful if there is no other data supporting such observations. In the revised version, we have shown that for common traits like age and sex, low p-values can be determined if the compared groups are distinct in their phenotype. The identified proteins are also well-known in the literature. This is less so for the COVID-19-related analysis, and we explain why these are closely related to the design of our study.

From our point of view, combinations of the following reasons could result in less significant p-values than the clinical studies.:

- 1) **Severity and symptoms at sampling:** We want to stress that the samples were collected from the general population, not a clinical setting. This means the studied phenotypes represent a random distribution of time points of infection, health status, and, according to the answers to the questionnaires, predominantly milder infections. Since we did not study patients who came to the hospital due to severe COVID-19, our subject will, by design, present a wide range of disease activities. We also don't know when and for how long the common symptoms, such as fever, coughing, and loss of taste or smell, lasted before the participants sampled themselves. Consequently, our study will be more heterogeneous regarding symptoms and disease status at sample collection. This will reduce the possibility of finding highly significant associations.
- 2) **Matching of phenotypes:** We judge the lack of novel biomarkers with flashier p-values as a good indication of the data quality, the integrity of the samples, and the careful matching of the subjects. We often observe a bias in published studies in which a group of symptom-free seronegative persons is compared with seropositive subjects with severe symptoms. The tables show that we included groups with congruent demographics and symptoms. Hence, the only known differences are limited to their serostatus. We, of course, acknowledge the limitation that we know very little else about donors. Thus, other hidden variables could have contributed to the lack of lower p-values, such as general lifestyle, current health or health history, and socioeconomic factors.
- 3) **Biomarker relevance:** Most studies on COVID-19 have investigated inflammatory proteins, which, by their intrinsic function, respond with noticeable changes in circulating levels. On the other hand, these markers are pleiotropic and often fluctuate due to other courses. Our focus had been to study more abundant proteins that we can measure reliably in this new approach. Hence, many included proteins would represent others, such as cardiometabolic phenotypes. From our selection, we shortlisted markers of immune response and innate immunity, demonstrating that these can be readily detected. We have discussed such limitations in the discussion.
- 4) **Power:** As discussed under the topic 'cohort size', a larger number of samples may be needed to reach better statistical power. This will be addressed in follow-up studies that go beyond the context of our proof-of-concept investigation.

7. Did you correlate the proteins changes over a time course - progression of COVID-19 infection and recovery?

>> We thank the reviewer for this highly relevant question, and we do not have data from consecutive samplings of specific donors who experienced the infection. To demonstrate the value of longitudinal analysis, we present data from a single donor who provided samples on five consecutive occasions, starting two weeks after diagnosis. The shown trends illustrate changes in protein levels during the recovery phase.

We added a figure to the supplementary, showing examples of how some of the mentioned proteins change over time and in relation to anti-S levels of IgG and IgM. Due to this being an N=1 study, there is an intrinsic lack of statistical power, and we refrain from drawing further conclusions. Using the concept of the protein-protein correlation would require data from several donors. We, unfortunately, do not have the data to perform such an analysis.

8. PEA is limited to a few hundred target proteins and biased by the panel selection.

>> We agree with the reviewer. The limitation of the method is to study predefined proteins. This concern has been addressed in the discussion, where we state: "..., since the chosen method was built on pre-selected panels of proteins, we could have missed some relevant metabolic and inflammatory markers described in the current COVID-19 literature...."

9. I could not find the data for the longitudinal analysis (5 separate occasions during weeks 2 to 5 after symptom onset), which is mentioned in the method.

>> We thank the reviewer for the comments. We have added the figure to the supplementary. Due to this being an N=1 study, there is a lack of statistical power, and we refrain from drawing further clinical or biological conclusions. See the answer to question # 7.

Minor:

1. Fig.2. FDR<0.01 (*horizontal* dotted line)
2. Dataset S2, column J is not necessary. Column for "Interplate – Median CV %" does not exist.
3. Method section, convert 3000rpm to rcf.

>> We thank the reviewer for noticing these errors. The requested changes have been made using the updated data .

Reviewer #2 (Remarks to the Author):

This is an interesting study, performed in several parts. First, the authors performed a comparison of Olink's Proximity Extension Assay (PEA -- multiplexed ELISA with PCR readout) for 92 proteins in Dried Blood spots (DBS) with blood samples from the same individuals, using DBS sampling kits designed by the authors (a collection device using a fixed volume of blood). As in an earlier study (ref 15), good correlation between the two datasets were found, so the authors continued with large-scale study of DBS on blood samples collected in 2020 from possible COVID-affected patients. These samples were then analyzed by Olink who performed PEA using a panel of 276 proteins (Olink's cardiovascular III and cardiometabolomic panels).

I was not familiar with this PEA technique, but Olink claims that their 2-antibody per protein, with DNA tags to enable PCR amplification of the signal, eliminates the cross-reactivity observed in standard

ELISA. The use of panels targeting proteins in specific pathways would make this technique a targeted “semi-quantitative technique” (as stated on page 20 of the manuscript).

>> We thank the reviewer for a nice and concise summary of the methodology applied in our study. Our laboratory is certified and equipped with the instrumentation required to perform PEA analysis using kits and reagents provided by the company Olink.

Presumably, these panels of proteins were selected because the pathways were already known to be associated with COVID. Interestingly, different panels of proteins could distinguish between 4 classes according to different immune responses (IgM-IgG-, IgM+IgG+, IgM+IgG- and IgM-IgG+) (page 9). I am more familiar with mass-spectrometry-based biomarker discovery -- and the rather low percentage of potential biomarkers that actually successfully pass the validation step. Therefore, I am curious as to whether the authors intend to validate any of the potential biomarker proteins that they uncovered, and if so, how would this validation be done?

>> We thank the reviewer for the comment. As discussed in our manuscript, the investigated proteins were already combined into panels by the provider of the kits (Olink). Our initial criteria for selecting these panels was to ensure we could detect many proteins. Hence, we did not target low-abundant analytes, such as cytokines.

We agree with the reviewer’s description of our study and are thankful for pointing out this important aspect in biomarker studies. We are investigating the shortlisted proteins further and believe that the community will focus on well-studied markers, such as MBL2, to be analyzed in larger populations.

Over the last few years, the proteomics community has understood the value and importance of orthogonal validation. This implies using an alternative technology to perform technical validation of a candidate biomarker (e.g. MS to validate affinity-based assays). In our current setting, however, we would prefer to apply the quantitative measurements that allow for the analysis of larger sample sets and avoid switching assay formats. Quantitative versions of the PEA assays exist, and using protein concentration values will assist in comparing different studies. Here we need to assume that there is no bias between the chosen methods, which can occur by targeting different epitopes and/or prototypic peptides.

I was also a bit confused by the discussion of protein “pairs”, and how these pairs were selected from all of the up or down-regulated proteins (page 12). Perhaps a bit more explanation would be helpful.

>> We agree with the reviewer that the topic around protein pairs has been difficult to grasp. As explained below, we have removed this analysis from the new version to narrow the scope to findings with prior knowledge. Due to the new sample set (study 3), the results section has been revised.

Even though the differential correlation analysis provides new insights into coordinated changes in protein levels, there is too little external data to confirm the changes in correlation that we have observed. In addition, differential protein-protein correlation analysis of 260 features sums up to 33,780 unique tests. This burdens any interpretation of the obtained p-values. As shown in the qq-plots below, the obtained p-values did not deviate from the projected increase in p-value significance.

Given that our work focused on confirming previous knowledge in DBS samples and acknowledging the relatively small size of our study sets, we do not believe the differential correlation analysis adds necessary information to the manuscript.

Although the paper in general is well written, it could benefit from a bit of language-polishing by a native speaker. (Some examples are “It is worth noticing” → It is worth noting (line 282) and “Hard breath” → Difficulty breathing, on page 28). To assist the authors, I have attached an edited word file.

>> We thank the reviewer for the time in suggesting these edits. These have been taken care of in the new version.

Coming from a mass-spectrometry background, and knowing the cost of a standard ELISA assay, I am curious about the cost of this Olink PEA analysis for 276 targeted proteins. I am also curious about the cost of Olink versus label-free LC/MS/MS for the same number of proteins detected. Or possibly a comparison of a limited number of Olink targets versus multiple-reaction monitoring MS for biomarker validation.

>> Discussing the costs of analysis is, of course, an important subject and highly dependent on the number of analytes (92 proteins per panel) and samples (88 samples per plate). Communicating costs is subjective, as added costs for the staff’s hands-on time and maintenance of the infrastructure should also be considered. A key difference in cost calculation between the MS and Olink is the costs for kits (higher for Olink than shotgun MS) versus the cost for instrumentation, maintenance, and co-financed usage (higher for MS than Olink).

We believe that any statement of costs will not be accurate as local costs might differ too much. We kindly refrain from providing a cost factor.

The amplification capability of PEA would certainly be an advantage (PEA literature claims fg/mL sensitivity), but most of the proteins described in this paper are of medium to high abundance (page 18, lines 436-437). As the authors note on page 18, there is a study that combines PEA with mass spectrometry (ref 43), but according to this reference only 35 proteins were compared between PEA, data-dependent and data-independent LC/MS/MS methods. As the authors note on page 18, this is an important area for future research.

>> We thank the reviewer for raising this important topic. As the study by Petrera et al. shows, comparing proteins between platforms is not always possible because of the differences in analytical sensitivity between different methods. The community has started to accept PEA as a complementary method to MS since it allows for analyzing low-abundant proteins. Our study focused on medium-abundant proteins to ensure a high coverage (= detecting proteins in many samples).

The DBS eluates represent a dilution of the 10 μ l of blood. Detecting even lower abundant proteins with the Olink technology may require using less diluted samples. We refrained from using the available volumes for such analyses.

On page 21, we wrote: *“Based on the assumption that 10 μ l of whole blood contains 50-60% fluid, meaning 5-6 μ l of plasma, and that we prepared eluates from a starting volume of 100 μ l elution buffer, we estimated that our eluates correspond to a plasma sample diluted 1:20. Eluates were diluted at 1:5, 1:1, or 1:101, respectively.”*

In summary, this is an interesting paper, but PEA is only semi-quantitative, and validation of the results presented here has not been addressed. I also have the feeling that PEA is probably very expensive (2 antibodies per protein, with DNA tags and PCR readout) -- and it seems to be only offered by Olink as a service. This would seem to limit its applicability.

>> We thank the reviewer for the positive assessment of our work. We want to redirect the reviewer's attention to the objective of our study: Confirming the possibility of studying clinically relevant proteins in DBS samples obtained from the general population.

Costs may differ for different users of a technology. Our lab has been certified for performing Olink assays since 2017. Hence, the raised concerns may not be limitations for everyone. In fact, a growing number of service sites and using ready-made kits make it straightforward for others to replicate our findings.

Many parallel efforts are ongoing in the community to compare data from Olink, MS, and other platforms. The near future will reveal which proteins will correlate between all these. Such work is beyond the scope of the presented study. The following review could offer some guidance on this matter:
<https://www.ncbi.nlm.nih.gov/pmc/articles/PMC9469506/>

Reviewers' comments:

Reviewer #2 (Remarks to the Author):

In this revised version, the authors increased the size of the study population (1228 individuals and the size of the panel to 276 proteins), and they have also addressed most of my questions.

In answer to Reviewer1's question #2 (i.e., "the findings do not reveal much new information") the previous version, could this study be considered to be a validation study of the earlier work, but one that utilizes a version of an ELISA assay that is less susceptible to cross reactivity? As has been pointed out before by other authors, a large panel of target proteins can be used as a biomarker discovery tool, as well as a validation method.

The authors have also demonstrates that, for the target proteins selected, DBS sample collection can be successfully used, which is also an important conclusion from this study.

I do, however, think that when a new method is being introduced, it is important to show the advantages and disadvantages of the new technique compared to existing methods so that researchers can have a basis for selecting the new method (PEA) versus the older ones (ELISA and MRM).

This includes a comparison of the precision, accuracy, reproducibility (%CVs) as well as the costs. These costs include the cost of having the assays done by a contract laboratory, or (alternatively) the cost of the instrumentation to have it done in one's own laboratory, the costs of available kits for various existing panels, and the cost (and time required) to add a new target protein.

The paper as a whole is much improved compared to the previous version. There also are only a few minor edits, which I have listed below:

Line 36. Add a reference to the PEA assay (for example,
<https://olink.com/content/uploads/2023/08/white-paper-pea-exceptional-specificity-v1.0.pdf>

Lines 43-44. early-inflection

Line 118-119. during May 2021

Line 172. Within a given sample type

Line 174. Finally, we investigated

Liine 187. Between the two sample types.

Line 198. We could only track all individuals back to the time of infection in one...

Line 199. For the other, we used only IgM and

Line 202. home-sampling

Line 203. Using their serostatuses,

Line 209 .. which we determined by detecting

Line 263. We found that eight samples deviated (Fig. S1), thus resulting in their exclusion from the summary tables

Line 293. Interestingly, these twelve proteins originated

Line 328. meta analysis identified age-associated proteins (? Missing word) in all datasets for the secreted

Line 368. Finally, we compared

Line 401. It is worth noting

Line 402. over a long period after the start of the pandemic (is this what was meant? Or do you mean that they occurred over a longer period of time than the shorter timeframe in which COVID started)

Line 405. Finally, we used

Line 412. no interactions have yet been reported

Line 421. They allow monitoring

Line 436. serostatuses

Line 437. On the proteome level.

Line 454. variance of the first principal

Line 469. participants got infected, on average, approximately five months before

Line 481. less-coordinated

Line 482. less-distinct were to be expected.

Line 487. we obtained samples from a single

Line 488. During the recovery phase Measured in duplicate, the

Line 489. COVID-induced

Line 490. protein-levels change between

Line 491. the time of sampling

Line 497. pathology. In addition to the proteins highlighted above,

Line 503. We also found SDCA to be a protein

Line 506 less-acute

Line 510. successfully used PEA

Line 516. blood-cell counts

Line 524. detected above the LOD as in a cell-free

Line 528. efforts will aim to improve

Line 602. were used to perform PEAs

Reviewer #3 (Remarks to the Author):

The authors have sufficiently addressed the reviewer's comments.

Dear Andreia Cunha and editorial colleagues at Communications Medicine

We are grateful to the reviewers and their positive assessment of our work. The remaining comments from Reviewer #2 have been addressed accordingly. We hope this revised version will be considered suitable for publication in Communications Medicine.

With Kind Regards,

Jochen Schwenk and co-authors.

Reviewer #2 (Remarks to the Author):

In this revised version, the authors increased the size of the study population (1228 individuals and the size of the panel to 276 proteins), and they have also addressed most of my questions.

In answer to Reviewer1's question #2 (i.e., "the findings do not reveal much new information") the previous version, could this study be considered to be a validation study of the earlier work, but one that utilizes a version of an ELISA assay that is less susceptible to cross reactivity? As has been pointed out before by other authors, a large panel of target proteins can be used as a biomarker discovery tool, as well as a validation method.

We thank the reviewer for the positive comment about our work. We agree that the aim here was to use Olink's PEA as a verification method with self-sampled DBS to target proteins previously identified in association with COVID-19 in samples collected in the clinic.

Olink's Target 96 panels are proximity extension assays that are more specific than classical ELISAs. The PEA strategy is to use antibodies coupled to complementary oligonucleotides so that protein detection only occurs when two antibodies bind to a common target protein. Once in close proximity to one another, the antibody-bound oligonucleotides can be hybridized to generate a template for real-time PCR amplification.

Cross-reactivity is, therefore, further reduced by the need for binding in proximity and by only two complementary oligonucleotides, leading to detectable PCR products. Therefore, from the point of technical validation to confirm previously reported biomarkers, PEA can be used as a discovery and verification method. We acknowledge that biomarker validation studies focus instead on clinically applicable specificity and sensitivity biomarker measurements, which require quantitative methods and certified standards.

The authors have also demonstrates that, for the target proteins selected, DBS sample collection can be successfully used, which is also an important conclusion from this study.

I do, however, think that when a new method is being introduced, it is important to show the advantages and disadvantages of the new technique compared to existing methods so that researchers can have a basis for selecting the new method (PEA) versus the older ones (ELISA and MRM).

This includes a comparison of the precision, accuracy, reproducibility (%CVs) as well as the costs. These

costs include the cost of having the assays done by a contract laboratory, or (alternatively) the cost of the instrumentation to have it done in one's own laboratory, the costs of available kits for various existing panels, and the cost (and time required) to add a new target protein.

We thank the reviewer for highlighting aspects of the PEA approach's analytical performances and costs and its relation to other methods used for biomarkers discovery, validation, and monitoring.

As we wrote in the results section (lines 267ff), reported in detail (see Data S2), and described in the discussion (lines 575ff), we have already very carefully assessed the reproducibility of protein detection in DBS using PEA as a semi-quantitative method (CV < 10%). In addition, we report associations to age, sex, and COVID-19 infections that match data from independent studies and blood sample types. Further to this, we have also observed good concordance between protein profiles in EDTA plasma collected by venous blood draw and DBS collected by finger pricking ($r_s > 0.7$). Consulting the supplier's validation data documents, we found support for the hemolysis tolerance for the assays: <https://olink.com/resources-support/document-download-center/>, meaning that the data from assays with eluted and diluted samples are likely not influenced by interferences from residual hemolysis.

To demonstrate the method's quantitative linearity, we spiked DBS with known concentrations of human proteins (see below). Since PEA does not report quantitative values, we assessed accuracy as % of recovery on a spike-in experiment. Recovery was estimated at 75-99% and depended on the target of interest and elution buffer. As exemplified below, we prepared a serial dilution of recombinant human VEGFA (8-6000 pg/mL) and spiked these into plasma or DBS. The latter was eluted by testing an alkaline or detergent-containing buffer. Linear decreases in NPX levels can be observed for plasma and DBS samples. The detergent-containing buffer reveals lower NPX values due to the additional dilution set occurring from the elution. Such steps can be avoided for plasma samples or when using buffers that can be evaporated.

Figure 1- Dilution series of spiked in VEGF to test linearity and replicated analysis. We tested plasma (pink), DBS eluted with an alkaline buffer (green) and DBS eluted with a detergent buffer (petrol). The dotted lines indicate the reported LOD for the assay.

There, we also tested the stability of the eluate 50 days apart and achieved a high correlation ($\rho > 0.97$). The reproducibility between replicated discs loaded with DBS from the same donor was equally high ($\rho: 0.99$). The precision between the assay developed for plasma samples was still excellent, even if it was lower for plasma (CV=3.6%) than DBS (CV=9.1%).

Here, we again wish to clarify for the reviewer that the main scope of an Olink analysis of DBS is to perform a targeted proteomic analysis, which can become quantitative upon inclusion of standards for other proteins (now branded as Olink Target 48: <https://olink.com/products-services/target/48-cytokine-panel/>). Therefore, developing quantitative assays for each protein targeted by Olink's panels applied, and their analytical assessment would go beyond the scope of our work.

In the discussion, we have added:

*“DBS sampling has been tested with other affinity proteomics methods and targeted MS (50). The benefit of PEA includes the high-multiplex capacity and the excellent sensitivity (low pg/mL); nevertheless, when quantitative monitoring of biomarkers is needed, other methods such as targeted MS, ELISA, or other quantitative multiplex assay platforms, such as Luminex, Quanterix or MesoScale, would be preferable. According to a comparative analysis (51), intra and inter-assay precision of ELISA (CV 2-12%) or targeted MS (CV 5-11%) was comparable with the values we and others have reported for PEA in DBS samples (CV < 10%; **DataS2**).”*

Due to the differences and dynamic nature of pricing, discounts, volumes, and local costs, we want to refrain from adding this discussion to our manuscript. Still, and from a cost point of view, PEA has shown to be the most convenient technology for a discovery approach. Considering that these are estimates from 2023, the cost per sample offered by academic facilities is less than 100 USD per sample, so roughly 1 USD per data point. To our knowledge, MS is still the most expensive approach, ranging from 200-500 USD per sample, but possibly measuring > 300 proteins (PMID: 38015820). Cost per data point may decrease dramatically in MS when high deep protein coverage is achieved. However, a quantitative MRM assay may cost 200 USD per sample and single data point. Considering the limited sensitivity of MS, both targeted and untargeted, affinity proteomics quantitative assays, which are replacing ELISAs due to their multiplex capacity and improved sensitivity, are still the most convenient method for validation study and quantitative biomarkers monitoring in large cohorts of samples. An 80-plex quantitative assay costs > 100 USD (about 1.5 USD per data point), where the cost per sample decreases with the number of biomarkers included in the panel.

The paper as a whole is much improved compared to the previous version. There also are only a few minor edits, which I have listed below:

Thank you for a thorough revision of the text. We have revised our manuscript according to your suggestions or stated otherwise below.

Line 36. Add a reference to the PEA assay (for example, <https://olink.com/content/uploads/2023/08/white-paper-pea-exceptional-specificity-v1.0.pdf>

We await the editor's recommendation to adhere to the journal style regarding adding references to the abstract.

Lines 43-44. early-inflection

Line 118-119. during May 2021

Line 172. Within a given sample type

Line 174. Finally, we investigated

Line 187. Between the two sample types.

Line 198. We could only track all individuals back to the time of infection in one...

The sentence has been revised as follows:

"Since not all individuals were diagnosed by PCR or experienced symptoms from the infection, we had only self-reported information about a diagnosed infection in one of the studies."

Line 199. For the other, we used only IgM and

Line 202. home-sampling

Line 203. Using their serostatuses,

Line 209 .. which we determined by detecting

Line 263. We found that eight samples deviated (Fig. S1), thus resulting in their exclusion from the

summary tables

Line 293. Interestingly, these twelve proteins originated

Line 328. meta analysis identified age-associated proteins (? Missing word) in all datasets for the secreted

Line 368. Finally, we compared

Line 401. It is worth noting

Line 402. over a long period after the start of the pandemic (is this what was meant? Or do you mean that they occurred over a longer period of time than the shorter timeframe in which COVID started)

The sentence has been revised as follows:

"It is worth noting that, compared to study 1, infections of the seropositive participants in study 3 were not limited to the few months at the start of the pandemic."

Line 405. Finally, we used

Line 412. no interactions have yet been reported

Line 421. They allow monitoring

Line 436. serostatuses

Line 437. On the proteome level.

Line 454. variance of the first principal

Line 469. participants got infected, on average, approximately five months before

Line 481. less-coordinated

Line 482. less-distinct were to be expected.

Line 487. we obtained samples from a single

Line 488. During the recovery phase Measured in duplicate, the

Line 489. COVID-induced

Line 490. protein-levels change between

Line 491. the time of sampling

Line 497. pathology. In addition to the proteins highlighted above,

Line 503. We also found SDCA to be a protein

Line 506 less-acute

Line 510. successfully used PEA

Line 516. blood-cell counts

Line 524. detected above the LOD as in a cell-free

Line 528. efforts will aim to improve

Line 602. were used to perform PEAs

Reviewer #3 (Remarks to the Author):

The authors have sufficiently addressed the reviewer's comments.

We thank the reviewer for the positive assessment of our work.

REVIEWERS' COMMENTS:

Reviewer #2 (Remarks to the Author):

The revised manuscript is much improved, and it can be accepted for publication.

The authors have addressed most of my concerns, so I have only a few comments/suggestions left to make.

I found only a few awkward phrases, which the authors can fix later.

Line 560: Forward-looking, other recent studies --- Looking forward, other recent studies

Line 298: Despite being on separate panels, proteins such as LCN2, S100P, PAG1, and PLAUR were shown --- I'm not sure whether "on" or "in" is the correct preposition here. A person is on a panel, but I'm not sure what is the correct term for a protein!

Since PEA-Olink is new, should it be in the title or the abstract? (PEA is in the abstract, but not Olink)

I understand the authors' reluctance to discuss the cost of the PEA-Olink assay in the manuscript, and I don't want to delay publication of this paper, so perhaps they could discuss it in a different forum. (perhaps at a scientific meeting such as ASMS?).

For reference, a "standard" antibody costs ca. 3000 Euros and it takes 3 months to have them made. How does that compare with PEA antibody?

How does PEA-Olink (for a single protein target) compare with the cost of a standard ELISA for the same protein? (Obviously, PEA is designed to increase the specificity, but at what price?) Of course, you can use an antibody for multiple analyses. And after you have made the PEA antibody, how many assays can it be used for?

I agree that the cost of MS-based assays is currently a limiting factor for their use in large-scale projects, but this is an active area of research. Fractionation methods, typically needed because of the wide dynamic range of protein concentrations in plasma, increase the cost because of an increase in the number of samples, and depletion typically relies on antibodies.

A recently-reported method (April 2023) involving perchloric acid precipitation (<https://answers.childrenshospital.org/plasma-proteomics/>) claims to cost only \$2.50 per sample, and "enable(s) the detection of more than 1,500 proteins per sample at a rate of 60 samples per day". Of course, this is detection, and not quantitation, but if this is semi-quantitative, but PEA maybe semi-quantitative as well. As stated in the revised manuscript:

The benefit of PEA includes the high-multiplex capacity and the excellent sensitivity (low pg/mL); nevertheless, when quantitative monitoring of biomarkers is needed, other methods such as targeted MS, ELISA, or other quantitative multiplex assay platforms, such as Luminex, Quanterix or MesoScale, would be preferable.

MRM-MS with stable-isotope labeled standards, is a quantitative method, so comparing it to PEA-Olink may be comparing apples to oranges, but this is an interesting new technique and I am trying

to figure out its analytical “niche”.

Dear Andreia Cunha and editorial colleagues at Communications Medicine

We are excited to hear that you, in principle, accepted our manuscript for publication. The remaining comments from Reviewer #2 have been addressed accordingly. We have refrained from continuing the discussion around Olink as this goes beyond the scientific content.

After rearranging the sections based on the editorial requests and some final finetuning, we look forward to sharing our work with the community through Communications Medicine.

With kind regards,

Jochen Schwenk and co-authors.

The revised manuscript is much improved, and it can be accepted for publication.
> We thank the reviewer for the positive assessment of our work.

The authors have addressed most of my concerns, so I have only a few comments/suggestions left to make.

> We thank the reviewer for the positive assessment of our work.

I found only a few awkward phrases, which the authors can fix later.
Line 560: Forward-looking, other recent studies --- Looking forward, other recent studies

Line 298: Despite being on separate panels, proteins such as LCN2, S100P, PAG1, and PLAUR were shown --- I'm not sure whether "on" or "in" is the correct preposition here. A person is on a panel, but I'm not sure what is the correct term for a protein!

> We thank the reviewer for the comments, and we have revised our manuscript accordingly.

Since PEA-Olink is new, should it be in the title or the abstract? (PEA is in the abstract, but not Olink)

> We prefer to mention the assay, not the company providing the technology.

I understand the authors' reluctance to discuss the cost of the PEA-Olink assay in the manuscript, and I don't want to delay publication of this paper, so perhaps they could discuss it in a different forum. (perhaps at a scientific meeting such as ASMS?).

> We thank the reviewer for this comment and the ones below. We prefer to leave these out of our manuscript as they go beyond the scientific scope of our work.

We kindly invite the reviewer to consult with white papers provided by Olink (<https://olink.com/application-category/white-papers/>) and look out for the frequently reoccurring events presenting the PEA technology. We are users of this technology.

For reference, a "standard" antibody costs ca. 3000 Euros and it takes 3 months to have them made. How does that compare with PEA antibody?

How does PEA-Olink (for a single protein target) compare with the cost of a standard ELISA for the same protein? (Obviously, PEA is designed to increase the specificity, but at what price?) Of course, you can use an antibody for multiple analyses. And after you have made the PEA antibody, how many assays can it be used for?

I agree that the cost of MS-based assays is currently a limiting factor for their use in large-scale projects, but this is an active area of research. Fractionation methods, typically needed because of the wide dynamic range of protein concentrations in plasma, increase the cost because of an increase in the number of samples, and depletion typically relies on antibodies.

A recently-reported method (April 2023) involving perchloric acid precipitation (<https://answers.childrenshospital.org/plasma-proteomics/>) claims to cost only \$2.50 per sample, and "enable(s) the detection of more than 1,500 proteins per sample at a rate of 60 samples per day". Of course, this is detection, and not quantitation, but if this is semi-quantitative, but PEA maybe semi-quantitative as well. As stated in the revised manuscript:

The benefit of PEA includes the high-multiplex capacity and the excellent sensitivity (low pg/mL); nevertheless, when quantitative monitoring of biomarkers is needed, other methods such as targeted MS, ELISA, or other quantitative multiplex assay platforms, such as Luminex, Quanterix or MesoScale, would be preferable.

MRM-MS with stable-isotope labeled standards, is a quantitative method, so comparing it to PEA-Olink may be comparing apples to oranges, but this is an interesting new technique and I am trying to figure out its analytical "niche".